# CCD: Mitigating Hallucinations in Radiology MLLMs via Clinical Contrastive Decoding

## Abstract

Multimodal large language models (MLLMs) have recently achieved remarkable progress in radiology by integrating visual perception with natural language understanding. However, they often generate clinically unsupported descriptions, known as medical hallucinations, which pose serious risks in medical applications that demand accuracy and image-grounded outputs. Through empirical analysis, we find that prompt-induced hallucinations remain prevalent in radiology MLLMs, largely due to over-sensitivity to clinical sections. To address this, we introduce Clinical Contrastive Decoding (**CCD**), a *training-free* and *retrieval-free* inference framework that integrates structured clinical signals from task-specific radiology expert models. CCD introduces a dual-stage contrastive mechanism to refine token-level logits during generation, thereby enhancing clinical fidelity without modifying the base MLLM. Experiments on three datasets and multiple models demonstrate that CCD consistently improves overall performance on radiology report generation (RRG). On the MIMIC-CXR dataset, it yields up to a **2.78** absolute improvement in RadGraph-F1 when applied to state-of-the-art RRG models. Our approach provides a lightweight solution for mitigating medical hallucinations, effectively bridging expert models and MLLMs in radiology.

## 1 Introduction

Multimodal large language models (MLLMs) have recently shown substantial promise in the medical domain (AlSaad et al., 2024; Shen et al., 2025). By coupling vision encoders with pretrained large language models (LLMs) (Chen et al., 2024a; Liang et al., 2024), MLLMs align visual inputs with language representations (Liu et al., 2024b), enabling complex reasoning and generation across multimodal inputs (Yin et al., 2024; Liu et al., 2024a; Wang et al., 2024a). Among various medical specialties, radiology has emerged as a key application area (Tu et al., 2025; Saab et al., 2025), where MLLMs are increasingly used to interpret radiographs and articulate diagnostic findings in clinically precise language (Liu et al., 2019). Compared to general-domain settings, radiology imposes significantly stricter demands on factual accuracy and clinical reliability (Chen et al., 2024b).

Despite recent advances, MLLMs still face critical challenges that limit deployment in real-world settings, with hallucination being a primary concern (Huang et al., 2025). In clinical contexts, this issue is often termed *medical hallucination* (Chen et al., 2024c; Gu et al., 2024), referring to outputs that appear clinically plausible yet are unsupported by the medical image or misaligned with diagnostic intent (Zhu et al., 2025). Such errors are particularly consequential in safety-critical fields like radiology, where even minor inaccuracies can adversely affect diagnosis and ultimately compromise patient treatment (Chen et al., 2024b). In these scenarios, generated outputs must be grounded in medical evidence and adhere to established clinical standards (Wu et al., 2024).

Radiology report generation (RRG) involves automatically producing free-text reports from medical images (Liu et al., 2019), such as chest X-rays. As a core task in radiology workflows, it plays a central role in clinical interpretation and is a key benchmark for advancing medical AI (Monshi et al., 2020). Compared to visual question answering (VQA), which addresses narrowly scoped queries, RRG requires holistic image understanding and precise, clinically grounded expression of findings (Yildirim et al., 2024), making it substantially more complex and error-prone. Consequently, medical hallucinations in RRG are often more severe and multi-dimensional, including fabricated pathologies on normal images, misclassification of finding types or locations, and errors induced by

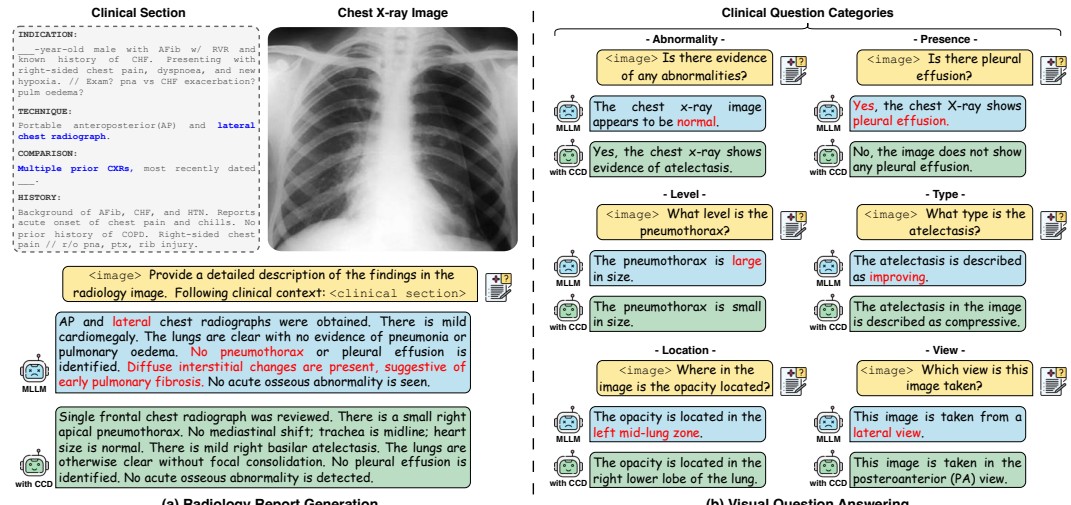

Figure 1: Illustration of the medical hallucinations in MLLMs across two tasks: **(a)** MAIRA-2 (Bannur et al., 2024) for the radiology report generation and **(b)** LLaVA-Med (Li et al., 2023a) for visual question answering. Medical hallucinations are highlighted in red, referring to generated clinical content that is not supported by the image. Clinically irrelevant or counterfactual information in the reference clinical section is shown in blue. With our Clinical Contrastive Decoding (CCD), medical hallucinations in the baseline models are mitigated across both tasks and question types.

contradictory prompts (Chen et al., 2024c), as in Figure 1 **(a)**. In contrast, hallucinations in VQA typically manifest as isolated factual inconsistencies (Zhu et al., 2025), as in Figure 1 **(b)**.

To mitigate medical hallucinations in RRG, recent advances have explored strategies such as restructuring training data (Zambrano Chaves et al., 2025), sanitising clinical sections using GPT-4V (OpenAI, 2024), and applying retrieval-augmented generation (RAG) (Xia et al., 2025; Hou et al., 2025). However, these approaches often raise privacy concerns, require costly retraining or access to proprietary APIs, and are impractical in low-resource radiology settings where constructing effective retrieval corpora is challenging. To investigate the persistence of medical hallucinations in radiology MLLMs, we conduct an empirical study on RRG in Section 3. Our findings reveal that prompt-induced medical hallucinations (Chen et al., 2024c), triggered by clinically implausible or ambiguous prompts, remain prevalent even when fine-grained inputs are provided (Figure 1, top-left). This highlights the need for inference-time solutions beyond dataset-level interventions.

Motivated by the aforementioned observations, we introduce **C**linical **C**ontrastive **D**ecoding (**CCD**), an inference-time method designed to mitigate medical hallucinations in radiology MLLMs. CCD adopts a two-stage hierarchical contrastive decoding framework that progressively incorporates external clinical signals to guide generation. Specifically, we leverage a task-specific expert model, such as a symptom classifier, to extract structured clinical labels and associated probabilities. Compared to the visual representations learned by the MLLM's vision encoder, the expert model provides more precise clinical information by capturing multiple symptom-level signals from the image. These signals are integrated in two complementary ways: predicted labels are injected as descriptive prompts to enhance the grounding ability of the MLLM, and probability scores are used to perturb the decoding process, both nudging the outputs toward clinical consistency. This framework enables MLLMs to benefit from additional image-derived knowledge without requiring further alignment or retraining. As a result, CCD is a *training-free* and *retrieval-free* approach that operates entirely at inference time to improve radiology MLLMs. This paper makes the following contributions [1]:

○ We conduct an empirical study on RRG and find that prompt-induced medical hallucinations remain prevalent in radiology MLLMs, often stemming from over-sensitivity to clinical sections.

○ We propose **CCD**, a general and lightweight inference-time framework that leverages radiology expert models to guide MLLM generation via structured labels and confidence-based guidance.

○ Extensive experiments across three datasets and multiple models show that **CCD** consistently enhances linguistic quality and clinical fidelity in RRG, while also improving accuracy on VQA.

---

[1] A detailed explanation of our research aim and scope is provided in Appendix A.1 and Appendix A.2.

## 2 RELATED WORK

**Radiology Multimodal Large Language Models.** Substantial advancements have been made in applying MLLMs to radiology, particularly for generating narrative-style reports directly from medical images (Sharma et al., 2024; Zhang et al., 2025c). This trend highlights the need for domain-specific MLLMs that can support clinical workflows, reduce the workload of radiologists, and improve patient care (Huang et al., 2023; Wu et al., 2023). Recent models such as Med-PaLM M (Tu et al., 2023), MAIRA-1 (Hyland et al., 2024), Lingshu (Team et al., 2025), and Med-Gemma (Sellergren et al., 2025a) have made encouraging progress. However, medical hallucination remains a key limitation, compromising the clinical reliability of MLLMs (Kim et al., 2025).

**Medical Hallucination in Multimodal Large Language Models.** Hallucination in LLMs is commonly defined as generating content that is irrelevant or unfaithful to the input (Tonmoy et al., 2024). In MLLMs, this often manifests as object hallucination, where generated outputs contradict the visual or factual evidence (Sahoo et al., 2024). Unlike general-domain applications, the medical domain presents unique triggers for hallucinations, such as clinically implausible prompts or subtle finding cues, and exhibits a markedly lower tolerance for errors (Wang et al., 2025b). The recent survey by Zhu et al. (2025) examines the causes of medical hallucinations and reviews current mitigation strategies. Among various contributing factors, strict privacy regulations exacerbate the scarcity and imbalance of clinical training data (Jiang et al., 2025a), which is a key cause of medical hallucinations and often more critical than factors introduced during training or inference (Hager et al., 2024). Corresponding mitigation strategies primarily focus on training-time interventions, such as constructing datasets that reflect a coherent chain of diagnostic reasoning Lai et al. (2025), followed by post-training (Banerjee et al., 2024) or deployment with RAG (Sun et al., 2025). At inference time, voting-based mechanisms have been adopted to improve accuracy in VQA (Liu et al., 2024c), but these approaches do not generalise well to the more complex RRG task.

**Radiology Report Generation.** RRG aims to generate free-text descriptions of clinical findings, establishing it as a central objective in automated medical imaging analysis (Wang et al., 2018). Recent efforts in RRG have primarily focused on improving the quantity and quality of training data to reduce medical hallucinations. LLaVA-Rad (Zambrano Chaves et al., 2025) uses an API-based model to sanitise noisy clinical sections, while retrieval-augmented generation has been explored to improve factual grounding (Li et al., 2024; Hou et al., 2025). Advanced models, MAIRA-2 (Bannur et al., 2024) integrates structured clinical sections and prior reports to improve diagnostic grounding, while Libra (Zhang et al., 2025c) mitigates temporal hallucinations by explicitly modelling historical image information. However, these approaches often require costly retraining, extensive dataset curation, and may raise privacy or security concerns. They also rely on retrieval infrastructure, which limits their practicality in out-of-distribution settings or when adapting to new benchmarks.

**Contrastive Decoding Strategies.** Contrastive decoding has emerged as an effective inference-time approach to mitigate hallucinations in generative models (Leng et al., 2023; Favero et al., 2024a), offering a lightweight alternative to costly training-time interventions. Visual Contrastive Decoding (VCD) (Leng et al., 2023) addresses object hallucinations by comparing output distributions between original and distorted visual inputs. Similarly, Instruction Contrastive Decoding (ICD) (Wang et al., 2024b) explores hallucination amplification under perturbed textual instructions. Alternative inference-time methods, such as VTI (Liu et al., 2024d), OPERA (Huang et al., 2024), M3ID (Favero et al., 2024b), and DeCo (Wang et al., 2025a), guide generation using shallow visual cues, fixed transformer layers, or token-level confidence scores. Recent work, such as Attn-Lens (Jiang et al., 2025b), achieves state-of-the-art performance in general-domain settings by integrating information across multiple attention heads. While effective in such domains, these methods struggle to mitigate medical hallucinations in radiology, partly due to the grayscale nature of imaging data and the scarcity of diverse, domain-specific datasets (Singhal et al., 2023). Moreover, radiology MLLMs are often trained for single tasks (e.g., RRG or VQA), which limits the generalisability of training-free strategies in clinical applications.

## 3 MEDICAL HALLUCINATION IN RADIOLOGY MLLMS

In this section, we conduct empirical analyses to examine the behaviour of radiology MLLMs and identify the causes of prompt-induced medical hallucinations (Chen et al., 2024c). Specifically,

Table 1: **Medical hallucination evaluation on MIMIC-CXR.** The baseline uses greedy decoding without clinical section input. "↑" indicates improvement; "↓" indicates degradation.

| Metric | Clinical Section | | | | | |
|---|---|---|---|---|---|---|
| | *w/o* | *w/* Indication | *w/* Technique | *w/* Comparison | *w/* History | *w/* All |
| **Lexical:** | | | | | | |
| ROUGE-L | 15.60 | 15.36 ↓0.24 | 15.61 ↑0.01 | 12.60 ↓3.00 | 15.64 ↑0.04 | 14.83 ↓0.77 |
| BLEU | 0.95 | 1.09 ↑0.14 | 0.98 ↑0.04 | 0.81 ↓0.14 | 1.07 ↑0.12 | 0.94 ↓0.01 |
| BERTScore | 38.19 | 36.05 ↓2.14 | 37.41 ↓1.05 | 30.07 ↓8.12 | 37.38 ↓0.81 | 35.53 ↓2.66 |
| **Clinical:** | | | | | | |
| RadGraph-F1 | 7.59 | 7.01 ↓0.58 | 7.35 ↓0.24 | 5.88 ↓1.71 | 7.53 ↓0.06 | 5.80 ↓1.79 |
| Temporal-F1 | 13.65 | 12.51 ↓1.14 | 12.97 ↓0.68 | 10.13 ↓3.52 | 13.11 ↓0.54 | 12.47 ↓1.18 |
| RaTEScore | 43.91 | 43.31 ↓0.61 | 43.78 ↓0.13 | 35.10 ↓8.81 | 43.74 ↓0.17 | 41.92 ↓1.99 |
| RadEval-BERT | 17.53 | 17.39 ↓0.14 | 17.07 ↓0.46 | 13.98 ↓3.57 | 17.39 ↓0.14 | 16.48 ↓1.05 |
| *CheXbert-F1 (Top5):* | | | | | | |
| Atelectasis | 43.07 | 37.51 ↓5.56 | 39.36 ↓3.71 | 31.29 ↓11.78 | 38.14 ↓4.93 | 22.17 ↓20.90 |
| Cardiomegaly | 7.49 | 14.39 ↑6.90 | 8.01 ↑0.52 | 6.29 ↓1.20 | 12.61 ↑5.12 | 11.45 ↑3.96 |
| Consolidation | 2.37 | 2.36 ↓0.01 | 2.25 ↓0.12 | 0.89 ↓1.48 | 0.78 ↓1.59 | 9.40 ↑7.03 |
| Edema | 11.59 | 15.11 ↑3.52 | 0.90 ↓10.69 | 2.67 ↓8.92 | 12.48 ↑0.89 | 19.19 ↑7.60 |
| Pleural Effusion | 54.24 | 48.38 ↓5.86 | 53.22 ↓1.02 | 41.84 ↓12.40 | 52.29 ↓1.95 | 43.18 ↓11.06 |

we focus on the chest X-ray modality and the RRG task, which requires comprehensive image understanding and is more susceptible to medical hallucinations than VQA. The quality of generated reports thus serves as a strong indicator of overall model performance. We conduct experiments on the widely used MIMIC-CXR dataset (Johnson et al., 2019b), whose detailed clinical sections provide a reliable reference for both evaluating hallucinations and guiding generation.

**Setup for Medical Hallucinations**  Prompt-induced hallucinations refer to errors triggered by prompts containing misleading or implausible information, thereby serving as a means to evaluate a model's robustness in clinically sensitive contexts (Chen et al., 2024c). Previous advanced work has primarily relied on incorporating clinical sections from radiology reports during MLLM training to enhance alignment (Bannur et al., 2024; Zhang et al., 2025c). However, such sections may contain irrelevant or invalid information. For instance, as illustrated in Figure 1 **(a)** (top-left), the clinical section references a *lateral view* and *prior CXRs*, which are counterfactual given that only a single frontal view is available. To assess such medical hallucinations, we prompt the model with varied clinical sections and evaluate whether it can robustly handle factual inconsistencies while maintaining the quality of the generated report. We choose LLaVA-Med v1.5 (Li et al., 2023a) as our baseline due to its extensive training with radiology visual instruction data and strong instruction-following capability. We adopt the default prompt shown in Figure 1 **(a)** and use greedy decoding, the standard setting for radiology MLLMs. In each case, we append a different clinical section, such as *indication*, *technique*, *comparison*, or *history*, to the end of the default prompt. These sections are extracted using rule-based heuristics from the MIMIC official repository (Johnson et al., 2018).

**Evaluation for Report Generation**  We follow prior work and adopt a set of lexical and radiology-specific metrics (Hyland et al., 2024; Zambrano Chaves et al., 2025), which are widely adopted as standard evaluation protocols in the field. Lexical metrics such as ROUGE-L (Lin, 2004), BLEU (Papineni et al., 2002), and BERTScore (Zhang et al., 2020) are used to measure textual overlap between generated and reference reports. For domain-specific evaluation, we employ a range of clinically grounded metrics. RadGraph-F1 (Delbrouck et al., 2022) evaluates overlap in clinical entities and relations. Temporal-F1 (Zhang et al., 2025c) measures the correctness of temporal descriptions (e.g., worsening or improvement). RaTeScore (Zhao et al., 2024) assesses the accuracy of medically relevant concepts such as anatomical structures and diagnoses. We also include RadEval-BERT (Xu et al., 2025a), a radiology-specific evaluation model trained on large-scale corpora to assess clinical semantic consistency. Finally, we use CheXbert-F1 (Smit et al., 2020) to assess the model's ability to accurately mention the five most common findings in generated reports (Irvin et al., 2019): Atelectasis, Cardiomegaly, Consolidation, Edema, and Pleural Effusion.

**Hallucination Drivers: Clinical Context Sensitivity.**  As shown in Table 1, appending different clinical sections leads to varying degrees of performance change. For lexical metrics, sections such as *history* and *technique* sometimes result in slight score improvements. This is because these

sections contain clinical terminology and standardised phrasing that resemble the narrative style of radiology reports, thereby making the generated text appear more fluent. In contrast, adding the *comparison* section consistently leads to lower scores (e.g., BERTScore ↓ 8.12). This is because comparison notes often include references to prior exams or temporal changes, which are not observable in the current frontal image. This mismatch between the textual prompt and the visual input introduces context that the model cannot validate, increasing the likelihood of hallucinated content.

For clinical evaluation metrics, we observe a general decline in report quality across all appended sections. Interestingly, when appending *indication*, there is a modest improvement in the detection of certain pathologies, particularly *Cardiomegaly* (CheXbert-F1 ↑ 6.90). This condition often co-occurs with other diseases and is frequently referenced in prior reports or diagnostic histories (Tavora et al., 2012), which may help the model retrieve relevant context during generation. Conversely, performance on findings such as *Pleural Effusion* and *Atelectasis* tends to decrease. These are typically late-stage manifestations (Woodring & Reed, 1996) that require fine-grained visual reasoning. When MLLMs place excessive emphasis on clinical textual guidance, they may overlook subtle visual evidence of pathological changes, leading to medical hallucinations. This suggests that such errors partly stem from the model's overreliance on prompt-injected clinical context.

Our empirical observations indicate that clinical sections in original reports are not always reliable sources of guidance for MLLMs during generation. In some cases, they introduce misleading signals that can adversely affect downstream tasks such as RRG. Therefore, selecting clinically relevant and contextually appropriate information is essential, particularly during inference. Motivated by this, our proposed CCD leverages domain-specific expert models to extract accurate and well-grounded clinical information, avoiding the ambiguity and noise often present in original report sections.

# 4 CLINICAL CONTRASTIVE DECODING

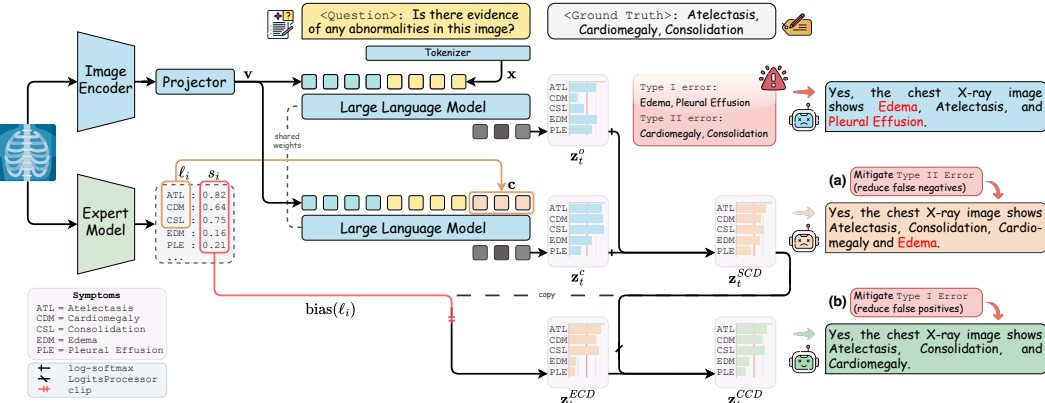

Figure 2: Overview of the CCD framework, which leverages a foundation expert model to enforce clinical consistency in MLLM outputs. During inference, it operates in two stages: **(a) Symptom-grounded Contrastive Decoding**, which incorporates structured clinical labels from the expert model; and **(b) Expert-informed Contrastive Decoding**, which adjusts the latent token logits using expert-derived confidence scores. The output logits are hierarchically calibrated to better match the ground-truth clinical labels. Hallucinated symptoms in the model output are marked in red.

As discussed in Section 3, radiology MLLMs tend to overreact to clinical context, leading to hallucinations that degrade report quality. To address this issue, we propose **C**linical **C**ontrastive **D**ecoding (**CCD**), a practical inference-time framework that dynamically adjusts token logits by incorporating clinically grounded signals from domain-specific expert models. As illustrated in Figure 2, CCD consists of two key stages: **(a)** Symptom-grounded Contrastive Decoding, which aligns the MLLM's self-perception with expert-derived symptom labels to reduce false negatives; and **(b)** Expert-informed Contrastive Decoding, which applies expert constraints to suppress false positives. Together, they mitigate both under-detection and over-diagnosis, improving clinical reliability.

**Preliminaries of MLLM Generation.** MLLMs are typically composed of a pretrained visual encoder, a language model as the text decoder, and a projection layer that maps visual tokens into the

latent space of the LLM. The projected visual tokens are dimensionally aligned with the embedded text tokens and then fed into the autoregressive language model for generation. For clarity, we denote the projected visual tokens as $\mathbf{v} = \{v_1, v_2, \ldots, v_n\}$, where each $v_i \in \mathbb{R}^d$ and $d$ is the hidden dimension. For the default prompt, we represent it as $\mathbf{x} = \{x_1, x_2, \ldots, x_m\}$, where each $x_j \in \mathbb{R}^d$ and $m$ is the number of textual tokens. Let $f_\theta$ denote the MLLM parameterized by $\theta$. Given the visual tokens $\mathbf{v}$ and textual tokens $\mathbf{x}$, the model generates a response sequence $\mathbf{y} = \{y_1, \ldots, y_T\}$, where each $y_t \in \mathcal{V}$ is a token from the vocabulary of the language model. Accordingly, the output logits at decoding step $t$ are denoted as $\mathbf{z}_t^o = f_\theta(\mathbf{v}, \mathbf{x}, y_{<t}) \in \mathbb{R}^{|\mathcal{V}|}$.

## 4.1 SYMPTOM-GROUNDED CONTRASTIVE DECODING (SCD)

SCD builds on the idea of contrastive decoding (Li et al., 2023b), which encourages generation that aligns with a target model while staying distinct from a constraint model. This approach balances fluency and factuality by comparing token likelihoods between models. In our setting, we adapt this framework to radiology by introducing symptom-level signals from a task-specific expert model, guiding the MLLM to avoid false negatives without retraining.

**Initial Anchor from Experts.** Given the diverse symptoms encountered in real-world clinical settings, we focus on the 14 pathology labels defined in the CheXpert ontology (Irvin et al., 2019) as our target set. To obtain symptom-level supervision, we use a DenseNet-based classifier [2] pre-trained on the MIMIC-CXR dataset (Johnson et al., 2019b) to predict the 14 pathologies from a given $\mathbf{v}$, which is widely used as a baseline in medical image classification (Baltruschat et al., 2019). From this expert model, we extract a set of clinical labels $\mathcal{L} = \{(\ell_i, s_i)\}_{i=1}^{M}$, where each $\ell_i$ denotes a finding (e.g., "Atelectasis"), and $s_i \in [0, 1]$ represents its predicted probability. These expert-provided symptom labels are filtered using a default threshold (e.g., $s_i > 0.5$), and the selected labels are then used to construct a concise anchor prompt (e.g., "Attention to the following clinical instructions: Atelectasis, Cardiomegaly, ..."), denoted as $\mathbf{c}$, which guides the model during generation.

**Self-perception Alignment.** The model generates its internal symptom representation by producing token-level logits conditioned on the initial clinical anchor. For the same image $\mathbf{v}$, this can be expressed as $\mathbf{z}_t^c = f_\theta(\mathbf{v}, \mathbf{x} \oplus \mathbf{c}, y_{<t}) \in \mathbb{R}^{|\mathcal{V}|}$, where $\oplus$ denotes concatenation. This design aims to guide the MLLM to generate more relevant symptoms by leveraging the additional clinical context, thereby reducing false negatives. We refer to this guided prediction path as the contrastive branch.

**Internal Guidance.** Following the analysis in Section 3, we note that excessive reliance on clinical context can also lead to hallucinations. To balance the influence of the contrastive branch ($\mathbf{z}_t^c$) and the original decoding branch ($\mathbf{z}_t^o$), we integrate them using a contrastive decoding mechanism. To ensure numerical stability and facilitate comparison between distributions from different inputs, we convert logits into log-probabilities using log-softmax:

$$\tilde{\mathbf{z}}_t^o = \log \mathrm{softmax}(\mathbf{z}_t^o), \qquad \tilde{\mathbf{z}}_t^c = \log \mathrm{softmax}(\mathbf{z}_t^c) \tag{1}$$

This transformation mitigates scale and shift sensitivity between outputs, especially when the initial anchor induces large deviations from the original distribution. It also prevents unintended amplification of non-symptom tokens. The generation of the $t$-th output token is then given by:

$$\mathbf{z}_t^{\mathrm{SCD}} = (1 - \alpha)\,\tilde{\mathbf{z}}_t^o + \alpha\,\tilde{\mathbf{z}}_t^c \tag{2}$$

where $\alpha \in [0, 1]$ balances original and anchor-conditioned logits. This encourages the model to align generation with clinically meaningful findings, serving as an internal contrastive signal. At this stage, false negatives are primarily suppressed, as illustrated in Figure 2 **(a)**.

## 4.2 EXPERT-INFORMED CONTRASTIVE DECODING (ECD)

Inspired by Bayesian conditional reasoning (Barber, 2012), ECD further incorporates expert model signals to guide the MLLM's generation process toward clinically plausible outputs.

**Probabilistic Guidance.** For each symptom $\ell_i$ with probability score $s_i$, we define a token-level bias using a logit transformation:

$$\mathrm{bias}(\ell_i) = \log \frac{s_i}{1 - s_i} \tag{3}$$

---

[2] By default, we use the DenseNet from TorchXRayVision (Cohen et al., 2021) for chest X-ray multi-label prediction. Section 5.3 presents an ablation study replacing it with MedSigLIP (Sellergren et al., 2025b).

Since these original probability scores $s_i$ reside in a different space from the MLLM's token logits $\mathbf{z}_t^o$, both in scale and semantics, they cannot be directly injected into the decoding stage of MLLMs. To address this, we transform them into token-aligned logit-based biases, ensuring compatibility with the model's output distribution and enabling smooth integration during inference.

**Diagnostic Plausibility Constraint.** Inspired by clinical practice, where likelihood ratios of 2, 5, and 10 are commonly interpreted as indicating weak, moderate, and severe diagnostic evidence, respectively (Deeks & Altman, 2004; Grimes & Schulz, 2005), we cap the logit-based bias as follows:

$$\text{bias}(\tilde{\ell}_i) \leftarrow \text{clip}(\text{bias}(\ell_i), -\text{max\_bias}, +\text{max\_bias}), \qquad \text{max\_bias} = \log(\gamma) \tag{4}$$

where $\gamma \in \{2, 5, 10\}$. We incorporate the clipped bias to refine the first-stage SCD signal:

$$\mathbf{z}_t^{\text{ECD}} = \mathbf{z}_t^{\text{SCD}} + \text{bias}(\tilde{\ell}_i) \tag{5}$$

where $\tilde{\ell}_i$ is a selected symptom label from the expert model, and its corresponding bias is uniformly added to the token logits. This constraint limits over-correction while preserving the generative flexibility of the MLLM. To avoid interfering with inherent decoding behaviour, we apply default decoding controllers on the first-stage SCD logits, as:

$$\tilde{\mathbf{z}}_t^{\text{SCD}} = \text{LogitsProcessor}(\mathbf{z}_t^{\text{SCD}}) \tag{6}$$

where LogitsProcessor() refers to a stack of standard decoding modules from the Transformers library (Wolf et al., 2020), including commonly used components such as repetition penalties, minimum length constraints, and decoding strategies like temperature scaling, greedy decoding, and beam search. These modules ensure stable and consistent generation behaviour across models.

**Sustained Contrastive Adjustment.** While the first-stage SCD encourages the model to generate more symptom-related content, it may also increase the risk of false positives. To mitigate this, we incorporate expert-informed constraints to suppress clinically unjustified symptoms. Finally, we interpolate between the adjusted SCD logits and the ECD output to produce the final token logits:

$$\mathbf{z}_t^{\text{CCD}} = (1 - \beta)\,\tilde{\mathbf{z}}_t^{\text{SCD}} + \beta\,\mathbf{z}_t^{\text{ECD}} \tag{7}$$

where $\beta \in [0, 1]$ balances the contributions of internal contrastive and expert-informed logits, preventing over-reliance on existing true positives while maintaining linguistic fluency. The final next-token distribution is computed as $p(\tilde{y}_t \mid \cdot) = \text{softmax}(\mathbf{z}_t^{\text{CCD}})$, where $\tilde{y}_t$ denotes the probability of the token generated at decoding step $t$ after dual-stage adjustment.

As illustrated in Figure 2 **(b)**, **CCD** integrates symptom-grounded and expert-informed signals to continuously adjust the MLLM's output during inference, refining the autoregressive decoding process and mitigating both false negatives and false positives in medical hallucinations.

## 5 EXPERIMENTS

In this section, we conduct a series of experiments to evaluate the effectiveness of CCD in mitigating medical hallucinations and improving performance in radiology-specific generation tasks. Our evaluation spans multiple radiology MLLMs, three datasets, and two key tasks: RRG and VQA.

### 5.1 EXPERIMENTAL SETTINGS

**Datasets.** We evaluate our method on three widely used radiology datasets: the official test splits of MIMIC-CXR (Johnson et al., 2019b) and IU-Xray (Demner-Fushman et al., 2015), and the public validation set of CheXpert Plus (Chambon et al., 2024), as no official test split is available for the latter. Following prior works (Sharma et al., 2024; Zhang et al., 2025c), we focus on generating the *findings* section from a single frontal-view image for the RRG. For the VQA task, we use Medical-CXR-VQA (Hu et al., 2024), a MIMIC-CXR-derived dataset with six clinical question categories, shown in Figure 1 **(b)**. Additional dataset details are provided in Appendix B.1.

**Evaluation Metrics.** We adopt the same set of metrics described in Section 3 to evaluate report generation quality. For the VQA task, we report micro-averaged Recall and F1 based on whether ground-truth labels appear in the generated text. For details on evaluation metrics, see Appendix B.2.

**Baselines.** In addition to the default greedy decoding strategy, we compare against several recent training-free hallucination mitigation methods proposed in the general domain, including VCD (Leng et al., 2023), OPERA (Huang et al., 2024), ICD (Wang et al., 2024b), DeCo (Wang et al., 2025a), and Attn-Lens (Jiang et al., 2025b). We primarily evaluate the effectiveness of our proposed CCD on two advanced radiology MLLMs: MAIRA-2 (Bannur et al., 2024) for RRG and LLaVA-Med (Li et al., 2023a) for VQA. We use the pathology classifier from TorchXRayVision (Cohen et al., 2021) as the expert model to provide symptom-level predictions from chest X-ray images. Additional decoding strategies and corresponding results are presented in Appendix D.1.

**Implementation Details.** For all methods, we adopt the default configurations from their original papers to ensure fairness. For CCD, we fix the hyperparameters across tasks: in the first stage, the symptom-grounded guidance strength is set to $\alpha = 0.5$; in the second stage, the expert-informed guidance strength is set to $\beta = 0.5$, and the diagnostic plausibility constraint is controlled by $\gamma = 10$. Additional details, including descriptions of MLLMs and expert model settings, are in Appendix C.

## 5.2 EXPERIMENTAL RESTULTS

Table 2: **Evaluation on the radiology report generation.** Results on the IU-Xray and CheXpert Plus datasets are reported only for our method. **Best** and second-best results are bolded and underlined, respectively. The △ row indicates the absolute score improvement over the baseline.

| Method | Lexical Metric | | | Clinical Metric | | | | | |
|---|---|---|---|---|---|---|---|---|---|
| | ROUGE-L | BLEU | BERTScore | RadGraph-F1 | Temporal-F1 | RaTEScore | RadEval-BERT | CheXbert$_{F1}^{5}$ | CheXbert$_{F1}^{14}$ |
| **MIMIC-CXR** | | | | | | | | | |
| **Baseline** | 19.57 | 1.61 | 49.56 | 16.23 | 12.11 | 50.82 | 16.96 | 16.14 | 10.57 |
| + VCD | 19.47 | 2.02 | 48.99 | 15.90 | 12.57 | 49.85 | 17.49 | 19.17 | 15.47 |
| + OPERA | 19.18 | 1.77 | 49.31 | 16.06 | 13.26 | 50.59 | 17.09 | 16.25 | 11.82 |
| + ICD | 17.43 | 2.02 | 46.58 | 13.65 | 13.98 | 47.01 | 17.13 | 17.25 | 12.26 |
| + DeCO | 19.40 | 1.65 | 49.33 | 15.93 | 12.95 | 50.65 | 17.27 | 16.60 | 11.57 |
| + Attn-Lens | 19.51 | 1.68 | 49.67 | 16.37 | 13.45 | 50.86 | 17.15 | 16.74 | 10.98 |
| + CCD | **20.70** | **2.10** | **51.62** | **19.01** | **17.58** | **53.32** | **17.50** | **27.05** | **16.02** |
| △ | 1.13 | 0.49 | 2.06 | 2.78 | 5.47 | 2.50 | 0.54 | 10.91 | 5.45 |
| **IU-Xray** | | | | | | | | | |
| **Baseline** | 18.50 | 2.67 | 42.19 | 16.52 | 66.06 | 46.86 | 20.15 | 4.02 | 24.14 |
| + CCD | **20.77** | **3.31** | **46.25** | **21.12** | **67.16** | **50.47** | **22.14** | **19.96** | **28.23** |
| △ | 2.27 | 0.64 | 4.06 | 4.60 | 1.10 | 3.61 | 1.99 | 15.94 | 4.09 |
| **CheXpert Plus** | | | | | | | | | |
| **Baseline** | 18.07 | 1.83 | 45.91 | 14.27 | 22.78 | 47.47 | 1.99 | 13.54 | 8.39 |
| + CCD | **18.59** | **1.84** | **46.64** | **14.89** | **32.04** | **47.55** | **2.91** | **14.76** | **9.75** |
| △ | 0.52 | 0.01 | 0.73 | 0.62 | 9.23 | 0.08 | 0.92 | 1.22 | 1.36 |

**Results on Radiology Report Generation** We use MAIRA-2 (Bannur et al., 2024), the top open-source model on the ReXRank leaderboard (Zhang et al., 2024), as our baseline. Table 2 shows that CCD consistently improves both lexical and clinical metrics. Appendix D provides additional comparisons with other methods (in Table 5) and reports results across different MLLMs (in Table 6). These results suggest that CCD consistently outperforms general-domain decoding strategies, especially on clinical metrics such as CheXbert$_{F1}^{5}$ (↑10.91) and RadGraph-F1 (↑2.78) on MIMIC-CXR. Furthermore, it enhances the performance of advanced radiology MLLMs on the RRG tasks.

Table 3: **Evaluation on the medical visual question answering.** "↑" indicates improvement, "↓" denotes degradation relative to the baseline. See Appendix F for analysis of the two degraded cases.

| Model | Question Classification | | | | | | | | | | | | | |
|---|---|---|---|---|---|---|---|---|---|---|---|---|---|---|
| | Abnormality | | Presence | | View | | Location | | Level | | Type | | Overall | |
| | F1 | Recall | F1 | Recall | F1 | Recall | F1 | Recall | F1 | Recall | F1 | Recall | F1 | Recall |
| **LLaVA-Med** | 35.06 | 21.25 | 77.72 | 63.55 | 39.93 | 24.95 | 10.73 | 5.67 | 3.84 | 1.96 | 10.64 | 5.62 | 41.49 | 26.17 |
| + CCD | 43.16↑ | 27.52↑ | 80.91↑ | 67.94↑ | 41.15↑ | 26.04↑ | 10.23↓ | 5.40↓ | 3.92↑ | 2.06↑ | 10.14↓ | 5.36↓ | 45.11↑ | 29.12↑ |

**Results on Visual Question Answering** We use LLaVA-Med v1.5 (Li et al., 2023a) as the baseline. As shown in Table 3, CCD leads to consistent improvements across most categories. A slight drop is observed for *Location* and *Type* questions, mainly due to the broader and more morphological nature of these findings (e.g., infiltrates, scarring), which are not well captured by the 14-category

expert model used for guidance. Nonetheless, CCD maintains competitive overall performance even in these cases, demonstrating robustness despite the absence of explicit morphological labels.

## 5.3 ABLATION STUDIES

As shown in Table 4, we conduct ablation studies on the RRG task using MAIRA-2 to assess the effectiveness of CCD under different configurations, guided by the following research questions.

Table 4: **Ablation studies of CCD.** "*w/o*" indicates removal of a component; "$\mapsto$" denotes replacement with an alternative. "$\uparrow$ / $\downarrow$" indicate performance change relative to the baseline.

| Method | Lexical Metric | | | Clinical Metric | | | | | |
|---|---|---|---|---|---|---|---|---|---|
| | ROUGE-L | BLEU | BERTScore | RadGraph-F1 | Temporal-F1 | RaTEScore | RadEval-BERT | CheXbert$_{F1}^{5}$ | CheXbert$_{F1}^{14}$ |
| **CCD** | 20.70 | 2.10 | 51.62 | 19.01 | 17.58 | 53.32 | 17.50 | 27.05 | 16.02 |
| *w/o* SCD | 18.22 $\downarrow$ | 1.26 $\downarrow$ | 49.40 $\downarrow$ | 16.71 $\downarrow$ | 13.81 $\downarrow$ | 51.59 $\downarrow$ | 16.65 $\downarrow$ | 19.02 $\downarrow$ | 12.06 $\downarrow$ |
| *w/o* ECD | 20.73 $\uparrow$ | 1.96 $\downarrow$ | 51.72 $\uparrow$ | 18.78 $\downarrow$ | 17.40 $\downarrow$ | 53.21 $\downarrow$ | 17.71 $\uparrow$ | 21.02 $\downarrow$ | 11.47 $\downarrow$ |
| *w/o* All | 19.57 $\downarrow$ | 1.61 $\downarrow$ | 49.56 $\downarrow$ | 16.23 $\downarrow$ | 12.11 $\downarrow$ | 50.82 $\downarrow$ | 16.96 $\downarrow$ | 16.14 $\downarrow$ | 10.57 $\downarrow$ |
| *All-class $\mapsto$ Top-5-class* | 20.98 $\uparrow$ | 1.95 $\downarrow$ | 51.89 $\uparrow$ | 19.27 $\uparrow$ | 17.99 $\uparrow$ | 53.27 $\downarrow$ | 17.78 $\uparrow$ | 26.78 $\downarrow$ | 14.34 $\downarrow$ |
| *DenseNet $\mapsto$ MedSigLIP* | 20.92 $\uparrow$ | 2.24 $\uparrow$ | 51.86 $\uparrow$ | 19.32 $\uparrow$ | 16.80 $\downarrow$ | 53.48 $\uparrow$ | 18.12 $\uparrow$ | 27.42 $\uparrow$ | 16.59 $\uparrow$ |

**Are both stages of CCD necessary for performance gains?** We evaluate the impact of removing either SCD or ECD. Excluding SCD, which addresses false negatives, leads to a notable decline in CheXbert$_{F1}^{5,14}$, indicating reduced coverage of symptom-related findings. In contrast, removing ECD causes a relatively smaller drop in clinical metrics compared to SCD, but slightly improves some lexical scores, suggesting its role in suppressing false positives and promoting concise, accurate descriptions. Eliminating both stages results in the most substantial overall degradation, confirming that SCD and ECD are complementary and jointly critical for mitigating medical hallucinations.

**Does CCD remain robust under different expert settings?** We evaluate the robustness of CCD by varying the expert model configurations, as shown in the last two rows of Table 4. Limiting the expert output to the top-5 most frequent symptoms slightly improves lexical and some clinical metrics, likely because a smaller label space reduces generation complexity. However, it leads to a larger drop in CheXbert$_{F1}^{14}$ ($\downarrow 1.68$) compared to CheXbert$_{F1}^{5}$ ($\downarrow 0.27$), underscoring the importance of maintaining broad label space coverage in the pretrained expert model. Replacing the default expert with MedSigLIP (Sellergren et al., 2025b), an open-source zero-shot symptom classifier introduced concurrently, yields consistent improvements across both metric types. These results indicate that CCD benefits from stronger expert guidance while remaining robust across different expert settings.

**What is the effect of guidance strength on generation?** We vary the control weights $\alpha$ and $\beta$, which modulate the influence of symptom-grounded signals and expert-informed confidence scores, respectively. These weights determine how much the expert model guides the radiology MLLM during generation. Figure 3 shows that the model achieves its best empirical RadGraph-F1 score when both guidance strengths reach 0.5, indicating the importance of balanced adjustment [3].

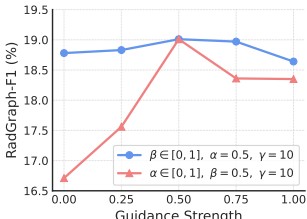

Figure 3: Ablation study of guidance strength ($\alpha$, $\beta$) ranging from 0 to 1, with others fixed at default.

## 6 CONCLUSION

In this work, we address the challenge of medical hallucinations in radiology MLLMs by introducing **C**linical **C**ontrastive **D**ecoding (**CCD**), a *training-free* and *retrieval-free* inference-time framework. By leveraging a task-specific expert model and dual-stage interventions on the MLLM's latent logits, CCD further improves clinical consistency in RRG and also contributes to VQA performance, all without retraining or data augmentation. Experiments across diverse models, datasets, and metrics validate its effectiveness in radiology tasks. Beyond performance, we highlight the complementary role of foundation expert models in guiding MLLM behaviour, offering a practical path to integrate domain expertise into generation models. As medical AI evolves, we believe CCD represents a modest yet meaningful step toward building more trustworthy and clinically aligned systems that approach physician-level reliability. A detailed discussion of limitations is provided in Appendix G.

---

[3]Appendix E includes detailed results, the ablation study of the plausibility constraint ($\gamma$), and random tests.

ETHICS STATEMENT

This study is conducted entirely using publicly available and de-identified datasets. We strictly adhere to the ethical guidelines and usage policies associated with each dataset, ensuring compliance with standards equivalent to CITI "Data or Specimens Only Research" certification or exempt human subjects research protocols. By relying exclusively on open-access data, we promote transparency, reproducibility, and ethical integrity in the development of AI systems. In all figures, the chest X-ray is blurred to preserve privacy and minimize visual discomfort.

The broader goal of this work is to support the development of medical AI systems that act as assistive tools for licensed clinicians rather than replacements. While such systems show strong potential for improving clinical efficiency and diagnostic accuracy, it is essential that they be deployed responsibly and with oversight from qualified radiologists to prevent unintended consequences. In particular, careful consideration is needed to avoid excessive reliance on automated outputs, which may reduce human involvement or worsen existing healthcare disparities. We promote a collaborative integration of AI and medical expertise to ensure that these technologies are used safely and equitably in clinical practice.

REPRODUCIBILITY STATEMENT

We are committed to ensuring the reproducibility of our results. Detailed descriptions of the model architecture, training configurations, and hyperparameters are provided in Section 5 and Appendix C. All datasets and baseline models used in our experiments are publicly available and can be accessed with the appropriate research-use certifications. Furthermore, the relevant source code has been included in the supplementary materials to facilitate replication of our experiments.

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

APPENDIX CONTENTS

# A    RESEARCH OBJECTIVES

## A.1    RESEARCH AIMS

This work introduces **C**linical **C**ontrastive **D**ecoding (**CCD**), a plug-and-play, inference-time framework designed to mitigate medical hallucinations in radiology multimodal large language models (MLLMs). The primary objective is to reduce clinically harmful errors, particularly prompt-induced hallucinations (Chen et al., 2024c), without modifying model parameters or requiring additional training. **CCD** enhances output reliability by integrating expert signals, such as predictions from pretrained pathology classifiers, during the decoding process. Designed to be model-agnostic, it applies broadly across MLLM architectures and tasks, including RRG and VQA.

To facilitate a fair comparison, it is also important to clarify what this work does not aim to address. We do not propose new model architectures or novel training methodologies. Our focus is on test-time decoding. Therefore, we do not compare with approaches that involve architectural modifications, additional training, or retrieval-based augmentation requiring external corpora. Nor do we attempt to eliminate all forms of medical hallucination. Instead, our focus is on reducing prompt-induced hallucinations that carry clinical importance or potential risk. Even the mitigation of a subset of hallucinations can lead to meaningful gains in overall task performance. For instance, in the case of view-type VQA tasks, symptom-guided decoding enables models to answer more accurately. This is because most findings are concentrated in frontal-view chest X-rays, whereas lateral-view images provide less diagnostic signal for common conditions (Bannur et al., 2024). As a result, incorporating expert-derived symptom likelihoods helps the model infer the appropriate view type, even when such information is not explicitly stated in the question.

## A.2    RESEARCH SCOPE

This study is restricted to the use of pretrained radiology-focused MLLMs for medical imaging tasks involving chest X-rays, which represent the most commonly used imaging modality in clinical practice. All experiments are conducted using only frontal-view chest radiographs, specifically anterior-posterior (AP) and posterior-anterior (PA) projections. We focus on two downstream tasks: radiology report generation (RRG) and visual question answering (VQA). The backbone models evaluated in this work include MAIRA-2 (Bannur et al., 2024), Libra (Zhang et al., 2025c), LLaVA-Rad (Zambrano Chaves et al., 2025), and LLaVA-Med (Li et al., 2023a). These models are used without any additional finetuning. For external guidance, we incorporate predictions from pretrained image-level expert models, either supervised classifiers (e.g., DenseNet from TorchXRayVision (Cohen et al., 2021)) or zero-shot vision-language models (e.g., MedSigLIP (Sellergren et al., 2025b)), that estimate the presence of clinical findings.

Several important areas are intentionally excluded from the scope of this work. We do not address other medical imaging modalities such as computed tomography (CT), magnetic resonance imaging (MRI), or ultrasound. Our framework does not incorporate multi-modality signals derived from clinical notes, laboratory values, or electronic health records (EHRs). Our scope is restricted to hallucinations arising in radiology-specific MLLMs, and does not extend to general-domain MLLMs. In particular, we focus on prompt-induced hallucinations, a critical and under-addressed subset of medical hallucinations. Furthermore, post-processing techniques such as output filtering, retrieval augmentation, or report rewriting are outside the focus of this study. The proposed **CCD** method operates entirely at inference time and does not require model retraining, which ensures compatibility with a wide range of pretrained models while maintaining low deployment overhead.

# B    DATASETS AND METRICS

## B.1    DATASETS DESCRIPTION

**MIMIC-CXR**    (Johnson et al., 2019b)    A large-scale, publicly available dataset comprising 377,110 chest radiographs from 227,835 imaging studies, each paired with a free-text radiology report. We make use of the JPEG images from the MIMIC-CXR-JPG release (Johnson et al., 2019a), which are derived from the original DICOM files. To ensure consistency, only anterior-posterior (AP) or posterior-anterior (PA) frontal views are retained.

Each report is preprocessed to extract five clinically relevant sections: *Findings*, *Indication*, *Technique*, *Comparison*, and *History*. This is done using pattern-matching heuristics based on the official preprocessing scripts (Johnson et al., 2018). We evaluate on the official test split, which consists of 2,461 studies that contain frontal-view images and non-empty "Findings" sections.

**IU-Xray** (Demner-Fushman et al., 2015)  A publicly available dataset for medical image analysis, consisting of 7,470 chest X-ray images and 3,955 corresponding diagnostic reports. To ensure compatibility with both MLLMs and expert models, all images are converted to PNG format. For evaluation, we select 3,307 frontal-view cases that include non-empty "Findings" sections.

**CheXpert Plus** (Chambon et al., 2024)  A large-scale dataset comprising 223,462 image–report pairs from 187,711 studies across 64,725 patients. Since the official test split is not publicly available, we use the validation set, which includes 72 frontal-view samples with non-empty "Findings" sections for evaluation on the report generation task.

**Medical-CXR-VQA** (Hu et al., 2024)  A large-scale visual question answering dataset derived from MIMIC-CXR, focusing exclusively on antero-posterior (AP) and postero-anterior (PA) chest X-ray views. It includes six predefined question types: *abnormality*, *location*, *type*, *level*, *view*, and *presence*. We use only the official test split, which contains 78,124 image–question pairs.

## B.2 EVALUATION METRICS

**Lexical Metrics**  We employ commonly used natural language metrics to assess the textual overlap between generated and reference reports. Specifically, ROUGE-L (Lin, 2004) measures the length of the longest common subsequence, BLEU (Papineni et al., 2002) computes n-gram precision with a brevity penalty, and BERTScore (Zhang et al., 2020) leverages contextual embeddings from BERT (Devlin et al., 2019) to assess semantic similarity. All metrics are computed with their default configurations. For BLEU, we report results using BLEU-4 (i.e., n=4), following prior work.

**Clinical Metrics**  We adopt several radiology-specific metrics to evaluate the clinical relevance and accuracy of generated reports. RadGraph-F1 (Delbrouck et al., 2022) parses reports into structured graphs composed of clinical entities (e.g., anatomical sites and observations) and their relations. Temporal-F1 (Zhang et al., 2025c) extends this by assessing the correctness of temporal descriptors such as "worsened," "improved," or "stable." RaTeScore (Zambrano Chaves et al., 2025) focuses on critical diagnostic concepts and anatomical details, offering robustness to medical synonyms and sensitivity to negation cues. RadEval-BERT (Xu et al., 2025a) leverages a radiology-adapted ModernBERT model (Warner et al., 2024) to assess semantic similarity between generated and reference reports. CheXbert-F1 (Smit et al., 2020) applies an automatic labeler to extract "present," "absent," or "uncertain" labels for 14 clinical conditions (Irvin et al., 2019); we report both the full 14-class F1 and the 5-class version for common pathologies.

To ensure fairness, reproducibility, and consistency with prior work, all lexical and clinical evaluation metrics are computed using the RadEval (Xu et al., 2025a) toolkit, with each metric applied using its default configuration.

**VQA Evaluation**  For the visual question answering (VQA) task, we report micro-averaged Recall and F1 scores, computed based on whether ground-truth labels are present in the generated responses. Since the model outputs are in free-form natural language (e.g., "There is evidence of opacity in the left lung."), and the ground truth is a structured label list (e.g., "atelectasis, opacity"), we only assess whether each reference label is mentioned in the generated text.

Specifically, true positives are counted as ground-truth labels that appear in the output, and false negatives are those that are missing. False positives are not penalised, as it is inherently difficult to determine which additional labels in a free-text sentence constitute hallucinations. This formulation aligns well with the clinical objective of ensuring that critical findings are not missed.

We adopt micro-averaging across all samples to reflect the overall coverage and correctness of label inclusion. Compared to macro-averaging, micro-averaging gives appropriate weight to frequent conditions and avoids over-penalising rare labels in sparse multi-label settings. This makes micro Recall and F1 the most suitable metrics for evaluating free-text VQA responses in radiology.

## C    EXPERIMENTAL DETAILS

In this section, we provide additional details about the four backbone MLLMs used in our experiments, along with the decoding strategies and expert model configurations. All experiments are conducted on two NVIDIA RTX 3090 GPUs (24GB memory each) with BF16 precision enabled. Since CCD is a fully test-time decoding strategy, it requires no additional training and can be applied directly to any pretrained MLLM. Despite incorporating an expert model and a two-stage decoding process, it maintains a lightweight deployment cost. On average, CCD incurs an inference-time overhead of approximately $1.45\times$ relative to standard greedy decoding. The actual runtime may vary depending on hardware configurations, particularly the floating-point operations per second (FLOPS) supported by the GPU.

### C.1    BACKBONE MODELS

**MAIRA-2** [4] (Bannur et al., 2024)    A model developed specifically for grounded radiology report generation, where the goal is not only to produce clinically accurate reports but also to localise findings within the image. The model is built upon the LLaVA framework (Liu et al., 2023), and incorporates a frozen Rad-DINO-MAIRA-2 vision encoder (Pérez-García et al., 2024), a Vicuna-7B (Chiang et al., 2023) language backbone, and a four-layer MLP that facilitates cross-modal alignment between image features and language representations.

**Libra**    (Zhang et al., 2025c)    A temporally-informed multimodal model designed for generating the *Findings* section in chest X-ray reports. Distinct from traditional single-image approaches, Libra processes longitudinal image pairs to capture disease evolution. It integrates a frozen Rad-DINO (Pérez-García et al., 2025) encoder with Meditron-7B (Chen et al., 2023), linked through a Temporal Alignment Connector. This connector incorporates a Layerwise Feature Extractor and a Temporal Fusion Module to encode multi-scale visual changes into a unified representation.

**LLaVA-Rad**    (Zambrano Chaves et al., 2025)    An instruction-tuned multimodal model designed for radiology report generation. It builds upon the LLaVA (Liu et al., 2023) architecture and employs LoRA (Hu et al., 2021) for parameter-efficient finetuning. To reduce training cost, the model is trained exclusively on MIMIC-CXR data, which offers high-quality radiology reports. These reports are further refined using GPT-4 (OpenAI, 2024) structuring to enhance label clarity and consistency. For visual encoding, LLaVA-Rad adopts a BiomedCLIP (Zhang et al., 2025b) model pretrained on biomedical image–text pairs, improving domain alignment with radiological content.

**LLaVA-Med**    (Li et al., 2023a)    A biomedical adaptation of the LLaVA (Liu et al., 2023) model, trained on a large-scale synthetic instruction-following dataset generated from PMC-15M (Zhang et al., 2025b) image–text pairs. Instructions are automatically generated using GPT-4 (OpenAI, 2024) without manual annotation. The model is finetuned in two stages: first aligning on biomedical image–text data, then learning open-ended instruction following. We use version 1.5 of LLaVA-Med, which adopts Mistral-7B (Jiang et al., 2023) as the language model and includes a jointly trained CLIP image encoder (Radford et al., 2021). This version is well-suited for biomedical VQA tasks, effectively handling clinical questions and extracting relevant findings from chest X-rays.

### C.2    METHOD CONFIGURATION

Since the MAIRA-2 (Bannur et al., 2024) model largely follows the LLaVA architecture (Liu et al., 2023), with the main differences being the use of a specialised image encoder and a four-layer fully connected multi-layer perceptron for vision-language alignment, we apply each training-free decoding method using the default LLaVA-type settings specified in its original publication. All comparison methods are implemented according to their published hyperparameter recommendations to enable fair and consistent evaluation. We do not perform any additional tuning of these hyperparameters beyond what is reported in the respective works. A summary of these decoding methods is provided in Appendix D.1.

---

[4]To ensure fair and consistent evaluation, chat templates and system prompts in MAIRA-2 are disabled; default instructions are provided to all models.

## C.3 Expert Model Setting

For the DenseNet model provided by TorchXRayVision (Cohen et al., 2021), we adopt the CheXpert Pathology Classifier, which is pretrained on the CheXpert dataset (Irvin et al., 2019). This model outputs probability scores for each of the 14 predefined pathologies, with label smoothing applied around the 0.5 threshold to enhance prediction stability. These confidence scores are directly used as expert guidance signals within our CCD framework.

For MedSigLIP (Sellergren et al., 2025b), a concurrent and publicly released variant of SigLIP (Zhai et al., 2023) tailored to encode medical images and text into a shared embedding space, we perform zero-shot classification over a predefined list of symptom labels following the official instruction format. Each prediction is based on a pair of textual prompts, such as "a chest X-ray with Atelectasis" and "a chest X-ray with no Atelectasis." By comparing the model's confidence scores for these alternatives, we obtain the probability associated with the positive prompt, which indicates the likelihood of the symptom being present in the image. These probabilities are subsequently used as expert-derived guidance signals in the CCD module.

# D Additional Experimental Results

## D.1 Comparison of Decoding Strategies on Radiology Report Generation

To provide a more comprehensive evaluation of CCD in comparison with other training-free hallucination mitigation methods, we expand upon the analysis in Section 5.2 by including an additional set of recent approaches. In total, we evaluate against eleven training-free methods under the same experimental settings. **The following is a brief overview of these methods.**

VCD (Leng et al., 2023) introduces contrastive decoding by comparing the output distributions from original and perturbed images. This approach reduces over-reliance on dataset priors and unimodal statistical biases. M3ID (Favero et al., 2024b) amplifies the influence of visual inputs during decoding, encouraging the model to generate tokens with higher visual-text mutual information. AVISC (Woo et al., 2025) detects visually misaligned tokens by examining attention patterns and dynamically refines the next-token prediction by contrasting logits from original versus visually-blinded inputs. OPERA (Huang et al., 2024) introduces a decoding-time penalty on logits to curb overconfidence, combined with a rollback mechanism that reviews earlier summary tokens and reallocates selections when needed. ICD (Wang et al., 2024b) contrasts the distributions from standard and instruction-perturbed inputs to amplify alignment uncertainty and effectively suppress hallucinated concepts embedded in the original distribution. PAI (Liu et al., 2024e) intervenes in the inference stage to steer the decoding process toward the original image perception direction, primarily by adjusting the self-attention heads in the decoder layers of MLLMs. VTI (Liu et al., 2024d) steers the latent space representations during inference to stabilise vision features, thereby reducing hallucinations. DeCo (Wang et al., 2025a) adaptively selects preceding layers and proportionally fuses their information into the final layer to dynamically adjust output logits. VISTA (Li et al., 2025b) mitigates hallucinations by combining two strategies: strengthening visual information in the activation space and utilising early-layer activations to guide more semantically coherent decoding. Attn-Lens (Jiang et al., 2025b) mitigates hallucinations by refining visual attention through the aggregation of signals from multiple attention heads. MARINE (Zhao et al., 2025) addresses object hallucinations by incorporating image-grounded guidance only at the prompt level into the decoding process. In our evaluation, we adopt the *MARINE-Truth* setting, using ground-truth labels of thoracic structures such as the lungs, heart, and pleural cavity as grounded references.

Additionally, in the general domain, numerous recent **training-free** methods have been proposed to mitigate hallucinations in MLLMs. These methods are publicly available and widely used within the research community. However, their underlying task assumptions are often incompatible with radiology-specific generation settings. For example, methods such as VDGD (Ghosh et al., 2025) first prompt an MLLM to generate a textual description of the image, which is then concatenated as a prefix to the original prompt. Similarly, SumGD (Min et al., 2025) constructs summarised instructions to guide the model prior to decoding. These types of strategies are not applicable to radiology models, which are often instruction-tuned for tasks such as radiology report generation. Since the report itself serves as a detailed image description, adding a separate generated caption will introduce redundancy or interfere with the model's instruction-following behaviour.

Table 5: **Comparison of report generation performance across decoding methods.** MAIRA-2 (Bannur et al., 2024), the top open-source model on the ReXrank (Zhang et al., 2024) leaderboard, is used as the baseline. Results on IU-Xray and CheXpert Plus are reported only for our method. **Best** and second-best results are bolded and underlined, respectively.

| Method | Lexical Metric | | | Clinical Metric | | | | | |
|---|---|---|---|---|---|---|---|---|---|
| | ROUGE-L | BLEU | BERTScore | RadGraph-F1 | Temporal-F1 | RaTEScore | RadEval-BERT | CheXbert$_{F1}^5$ | CheXbert$_{F1}^{14}$ |
| **MIMIC-CXR** | | | | | | | | | |
| **Baseline** | 19.57 | 1.61 | 49.56 | 16.23 | 12.11 | 50.82 | 16.96 | 16.14 | 10.57 |
| + *VCD* | 19.47 | 2.02 | 48.99 | 15.90 | 12.57 | 49.85 | 17.49 | 19.17 | 15.47 |
| + *M3ID* | 14.45 | 1.50 | 41.11 | 11.85 | 13.35 | 43.77 | 15.87 | 22.34 | 10.16 |
| + *AVISC* | 19.68 | 1.94 | 49.28 | 15.80 | 12.49 | 50.04 | 17.39 | 16.17 | 12.84 |
| + *OPERA* | 19.18 | 1.77 | 49.31 | 16.06 | 13.26 | 50.59 | 17.09 | 16.25 | 11.82 |
| + *ICD* | 17.43 | 2.02 | 46.58 | 13.65 | 13.98 | 47.01 | 17.13 | 17.25 | 12.26 |
| + *PAI* | 18.46 | 1.68 | 49.13 | 16.24 | 13.99 | 50.51 | 16.93 | 17.59 | 12.69 |
| + *VTI* | 19.21 | 1.68 | 49.77 | 16.42 | 13.48 | 51.20 | 16.87 | 12.13 | 8.75 |
| + *DeCO* | 19.40 | 1.65 | 49.33 | 15.93 | 12.95 | 50.65 | 17.27 | 16.60 | 11.57 |
| + *VISTA* | 10.98 | 0.80 | 36.59 | 6.43 | 13.61 | 38.94 | 16.84 | 26.28 | 15.82 |
| + *Attn-Lens* | 19.51 | 1.68 | 49.67 | 16.37 | 13.45 | 50.86 | 17.15 | 16.74 | 10.98 |
| + *MARINE* | 18.88 | 1.62 | 48.92 | 14.59 | 8.97 | 50.43 | 17.09 | 8.37 | 5.91 |
| + **CCD** | **20.70** | **2.10** | **51.62** | **19.01** | **17.58** | **53.32** | **17.50** | **27.05** | **16.02** |
| **IU-Xray** | | | | | | | | | |
| **Baseline** | 18.50 | 2.67 | 42.19 | 16.52 | 66.06 | 46.86 | 20.15 | 4.02 | 24.14 |
| + **CCD** | **20.77** | **3.31** | **46.25** | **21.12** | **67.16** | **50.47** | **22.14** | **19.96** | **28.23** |
| **CheXpert Plus** | | | | | | | | | |
| **Baseline** | 18.07 | 1.83 | 45.91 | 14.27 | 22.78 | 47.47 | 1.99 | 13.54 | 8.39 |
| + **CCD** | **18.59** | **1.84** | **46.64** | **14.89** | **32.04** | **47.55** | **2.91** | **14.76** | **9.75** |

While some methods, such as FarSight (Tang et al., 2025) and iTaD (Xu et al., 2025b), focus heavily on improving caption generation, their design motivations are largely driven by issues such as attention collapse, positional information decay, and the progressive reduction of attention weights to image tokens as model depth increases. However, these issues are less relevant for tasks such as visual question answering (VQA), which typically require only short, discrete responses. Consequently, such methods are not directly applicable to VQA settings.

Furthermore, some methods attempt to mitigate hallucinations by refining the visual input. For instance, ViCrop (Zhang et al., 2025a) performs automatic visual cropping to select important patch tokens, which are then re-concatenated with the original image tokens for generation. DyFo (Li et al., 2025a) leverages grounding-based visual expert models, such as Grounding DINO, to conduct visual search and eliminate object-level hallucinations. AGLA (An et al., 2025) uses adaptive masks to select relevant image patches as visual prompts, while masking out irrelevant regions. While these approaches have shown promising results in the general domain, their applicability to radiology is also limited. This is primarily due to the lack of strong pretrained grounding models in the medical domain, as well as the use of single-channel grayscale chest X-rays instead of three-channel natural images, which significantly constrains the applicability of visual prompt strategies in this setting.

In contrast to the methods discussed above, our proposed approach is more suitable for radiology MLLMs and the tasks defined within this setting. As shown in Table 5, the results reaffirm our earlier findings that CCD consistently improves the performance of backbone models across both lexical and clinical evaluation metrics. To further evaluate the clinical effectiveness of CCD, we additionally adopt the GREEN framework (Ostmeier et al., 2024) for both quantitative and qualitative assessment. GREEN leverages the natural language understanding capabilities of language models to identify and explain clinically significant errors in radiology reports. On MIMIC-CXR with the RRG task, the MAIRA-2 baseline achieves a GREEN score of 18.03. After applying CCD, the score increases to 19.14 (↑1.11), indicating better clinical alignment in the generated reports.

In summary, these results show that CCD is more effective for radiology-specific generation tasks than general-domain strategies, particularly for chest X-ray interpretation. This highlights its advantages in incorporating domain-specific knowledge into the decoding process.

## D.2 COMPARISON OF BACKBONE MLLMS FOR RADIOLOGY REPORT GENERATION

Table 6: **Overall performance on the radiology report generation task.** Our method is compared with baselines that use greedy decoding without any clinical section input. "↑" indicates improvement, "↓" denotes degradation relative to the baseline.

| Method | Lexical Metric | | | Clinical Metric | | | | | |
|---|---|---|---|---|---|---|---|---|---|
| | ROUGE-L | BLEU | BERTScore | RadGraph-F1 | Temporal-F1 | RaTEScore | RadEval-BERT | CheXbert$_{F1}^{5}$ | CheXbert$_{F1}^{14}$ |
| **MIMIC-CXR** | | | | | | | | | |
| LLaVA-Med | 15.60 | 0.95 | 38.19 | 7.59 | 13.65 | 43.91 | 17.53 | 25.78 | 21.89 |
| + CCD | 15.00 ↓ | 0.65 ↓ | 35.00 ↓ | 8.07 ↑ | 13.87 ↑ | 46.05 ↑ | 17.57 ↑ | 42.30 ↑ | 33.14 ↑ |
| LLaVA-Rad | 25.03 | 8.06 | 53.32 | 22.35 | 22.11 | 53.97 | 28.37 | 58.21 | 54.48 |
| + CCD | 25.32 ↑ | 7.43 ↓ | 54.24 ↑ | 23.52 ↑ | 22.59 ↑ | 55.70 ↑ | 28.30 ↓ | 58.22 ↑ | 54.63 ↑ |
| Libra | 21.50 | 4.74 | 50.52 | 20.46 | 19.59 | 53.13 | 24.99 | 59.46 | 51.76 |
| + CCD | 24.18 ↑ | 6.26 ↑ | 53.06 ↑ | 22.65 ↑ | 19.88 ↑ | 55.30 ↑ | 25.82 ↑ | 60.02 ↑ | 52.78 ↑ |
| **IU-Xray** | | | | | | | | | |
| LLaVA-Med | 11.94 | 0.39 | 34.58 | 7.14 | 60.23 | 43.02 | 20.12 | 7.71 | 5.44 |
| + CCD | 11.52 ↓ | 0.29 ↓ | 31.85 ↓ | 7.35 ↑ | 49.00 ↓ | 43.05 ↑ | 19.55 ↓ | 18.75 ↑ | 8.13 ↑ |
| LLaVA-Rad | 21.07 | 4.18 | 48.37 | 22.42 | 32.99 | 56.66 | 21.07 | 42.11 | 47.50 |
| + CCD | 25.36 ↑ | 5.62 ↑ | 56.38 ↑ | 31.73 ↑ | 36.80 ↑ | 64.94 ↑ | 23.24 ↑ | 42.48 ↑ | 47.56 ↑ |
| Libra | 24.31 | 2.99 | 51.59 | 26.38 | 59.06 | 56.22 | 23.63 | 43.86 | 45.46 |
| + CCD | 24.27 ↓ | 4.44 ↑ | 50.92 ↓ | 26.47 ↑ | 62.07 ↑ | 58.67 ↑ | 24.74 ↑ | 44.05 ↑ | 45.53 ↑ |
| **CheXpert Plus** | | | | | | | | | |
| LLaVA-Med | 14.40 | 0.72 | 32.59 | 4.63 | 25.00 | 42.16 | 4.63 | 25.84 | 25.00 |
| + CCD | 14.45 ↑ | 0.84 ↑ | 33.78 ↑ | 8.49 ↑ | 28.09 ↑ | 44.58 ↑ | 2.71 ↓ | 29.84 ↑ | 26.40 ↑ |
| LLaVA-Rad | 18.94 | 2.67 | 43.31 | 17.13 | 14.36 | 47.14 | 6.67 | 51.96 | 50.93 |
| + CCD | 19.43 ↑ | 2.66 ↓ | 47.16 ↑ | 17.81 ↑ | 23.89 ↑ | 50.31 ↑ | 6.73 ↑ | 51.99 ↑ | 51.37 ↑ |
| Libra | 18.87 | 2.14 | 47.04 | 19.20 | 27.18 | 49.33 | 7.58 | 45.68 | 50.08 |
| + CCD | 19.87 ↑ | 3.23 ↑ | 48.03 ↑ | 20.15 ↑ | 30.91 ↑ | 49.38 ↑ | 7.85 ↑ | 46.75 ↑ | 50.21 ↑ |

In addition to MAIRA-2 (Bannur et al., 2024) , we evaluate CCD on several other MLLMs to assess its generalisability in the radiology report generation task. These include Libra (Zhang et al., 2025c) and LLaVA-Rad (Zambrano Chaves et al., 2025), which are specifically tailored for the RRG task, as well as LLaVA-Med (Li et al., 2023a), a domain-specific foundation MLLM. We evaluate these models on three datasets: MIMIC-CXR (Johnson et al., 2019b), IU-Xray (Demner-Fushman et al., 2015), and CheXpert Plus (Chambon et al., 2024). Importantly, we do not tune the control strength hyperparameters of CCD. All models are evaluated using the default CCD settings, which may under-optimise performance for certain backbones.

As shown in Table 6, applying CCD consistently improves overall performance across all backbones, particularly in terms of clinical metrics. Interestingly, we observe that improvements in clinical consistency may occasionally come at the cost of lexical quality. For instance, LLaVA-Med exhibits a 1.64× gain in the CheXbert$_{F1}^{5}$, but also shows slight decreases in lexical metrics. This suggests that choosing appropriate hyperparameters for each model is critical to achieving a balanced trade-off between lexical and clinical performance. Overall, these results support the general applicability of CCD in enhancing radiology MLLMs across different architectures and evaluation settings, consistent with the conclusions drawn in Section 5.2.

## D.3 ALTERNATIVE BACKBONE MLLMS FOR VISUAL QUESTION ANSWERING

Following the same experimental settings as in Section 5.3 and Table 3, we further evaluate the generalisability of CCD using an alternative model, CheXagent-8B (Chen et al., 2024d). This model is an instruction-tuned foundation model for chest X-ray interpretation and integrates a vision encoder with a cross-modal adapter to align visual and textual representations.

Table 7: **Performance of CCD on the Medical Visual Question Answering with CheXagent-8B** "↑" indicates improvement, "↓" denotes degradation relative to the baseline.

| | Question Classification | | | | | | | | | | | | | |
| Model | Abnormality | | Presence | | View | | Location | | Level | | Type | | Overall | |
| | F1 | Recall | F1 | Recall | F1 | Recall | F1 | Recall | F1 | Recall | F1 | Recall | F1 | Recall |
| CheXagent | 61.75 | 45.74 | 68.18 | 59.40 | 38.28 | 23.97 | 7.67 | 3.99 | 27.24 | 15.77 | 8.54 | 4.46 | 47.67 | 33.12 |
| + CCD | 62.28 ↑ | 45.22 ↓ | 68.83 ↑ | 52.47 ↓ | 39.89 ↑ | 24.91 ↑ | 9.93 ↑ | 5.23 ↑ | 42.50 ↑ | 26.99 ↑ | 8.77 ↑ | 4.59 ↑ | 51.15 ↑ | 34.36 ↑ |

As shown in Table 7, CCD improves F1 scores across all evaluated categories and achieves better overall performance compared to the baseline. For specific categories such as *abnormality* and *presence*, we observe a moderate decrease in recall accompanied by a notable increase in precision, resulting in an overall gain in F1 score. This suggests that the model becomes more cautious in its predictions, producing fewer false positives while preserving overall reliability.

Moreover, employing a stronger backbone such as CheXagent-8B helps mitigate the modest declines observed in the *Location* and *Type* categories (see Appendix F), suggesting that improved base model capacity can complement CCD's effectiveness across question types. This is particularly beneficial in cases where expert models offer limited coverage of fine-grained radiological signals, such as lesion morphology or spatial localisation, which may otherwise bias the guidance process.

# E ADDITIONAL ABLATION STUDIES

## E.1 IMPACT OF VARYING CONTROL STRENGTH IN CLINICAL CONTRASTIVE DECODING

To understand the impact of guidance strength in **CCD**, we perform ablation studies by varying its three control hyperparameters ($\alpha$, $\beta$, and $\gamma$). In each experiment, we vary one hyperparameter while keeping the other two fixed, allowing us to isolate its effect on generation performance. These hyperparameters regulate the balance between the original MLLM output and the guidance from the clinical expert, determining how much influence each component has on the final generation. All experiments are conducted using MAIRA-2 (Bannur et al., 2024) as the backbone model, evaluated on the MIMIC-CXR (Johnson et al., 2019b) dataset for the radiology report generation task.

Table 8: **Ablation study of the $\alpha$ hyperparameter.** $\beta = 0.5$ and $\gamma = 10$ are used as default values. **Best** and second-best results are bolded and underlined, respectively. $\alpha \in [0, 1]$.

| $\alpha$ | Lexical Metric | | | Clinical Metric | | | | | |
| | ROUGE-L | BLEU | BERTScore | RadGraph-F1 | Temporal-F1 | RaTEScore | RadEval-BERT | CheXbert$^5_{F1}$ | CheXbert$^{14}_{F1}$ |
| --- | --- | --- | --- | --- | --- | --- | --- | --- | --- |
| 0.00 | 18.22 | 1.26 | 49.40 | 16.71 | 13.81 | 51.59 | 16.65 | 19.02 | 12.06 |
| 0.25 | 19.71 | 1.49 | 50.73 | 17.56 | 15.49 | 52.68 | 16.95 | 16.89 | 10.52 |
| 0.50 | 20.70 | 2.10 | 51.62 | **19.01** | **17.58** | **53.32** | 17.50 | 27.05 | 16.02 |
| 0.75 | 20.89 | 2.59 | 51.80 | 18.36 | 17.53 | 53.03 | **18.36** | 33.00 | **21.36** |
| 1.00 | **20.95** | **2.94** | **51.69** | 18.45 | 17.12 | 52.82 | 18.25 | **33.54** | 17.73 |

**Effect of $\alpha$ on Guidance Strength** As shown in Table 8, we investigate the effect of varying $\alpha$, which controls the overall guidance strength in the first stage of Symptom-grounded Contrastive Decoding. Increasing $\alpha$ strengthens the model's reliance on labels provided by the expert model to suppress false negatives. We observe that as $\alpha$ increases from 0 to 1, both lexical metrics and CheXbert-based scores consistently improve. However, other metrics such as RadGraph-F1 and RaTEScore begin to degrade once $\alpha$ exceeds 0.5.

This suggests that while stronger anchor label guidance can enhance entity coverage and clinical consistency, it may also result in overly verbose generations. Specifically, setting $\alpha = 1$ causes the model to fully rely on the initial expert-provided anchor, producing detailed descriptions that include more symptom labels and semantic content than necessary. To balance lexical fluency and clinical accuracy, we adopt $\alpha = 0.5$ as the default setting.

Table 9: **Ablation study of the $\beta$ hyperparameter.** $\alpha = 0.5$ and $\gamma = 10$ are used as default values. **Best** and second-best results are bolded and underlined, respectively. $\beta \in [0, 1]$.

| $\beta$ | Lexical Metric | | | Clinical Metric | | | | | |
|---|---|---|---|---|---|---|---|---|---|
| | ROUGE-L | BLEU | BERTScore | RadGraph-F1 | Temporal-F1 | RaTEScore | RadEval-BERT | CheXbert$_{F1}^5$ | CheXbert$_{F1}^{14}$ |
| 0.00 | **20.73** | 1.96 | **51.72** | 18.78 | 17.40 | 53.21 | **17.71** | 21.02 | 11.47 |
| 0.25 | 20.72 | 2.02 | 51.65 | 18.83 | 17.54 | 53.30 | 17.54 | 22.68 | 12.69 |
| 0.50 | 20.70 | **2.10** | 51.62 | **19.01** | **17.58** | **53.32** | 17.50 | 27.05 | 16.02 |
| 0.75 | 20.51 | 2.05 | 51.43 | 18.97 | 17.18 | 53.23 | 17.53 | **28.42** | 17.95 |
| 1.00 | 19.85 | 2.07 | 50.83 | 18.64 | 16.37 | 53.03 | 17.47 | 28.15 | **19.65** |

**Effect of $\beta$ on Guidance Strength**    As shown in Table 9, we investigate the effect of varying $\beta$, which controls the overall guidance strength in the second stage of Expert-informed Contrastive Decoding. Increasing $\beta$ corresponds to stronger reliance on the expert model's confidence scores, aiming to reduce false positives. We observe that as $\beta$ increases, clinical metrics, especially the CheXbert-based scores, consistently improve. However, lexical scores follow the opposite trend and gradually decrease. In addition, RadGraph-F1, Temporal-F1, and RaTEScore begin to decline when $\beta$ exceeds 0.5.

This degradation in lexical metrics is attributed to the model overfocusing on symptom-related descriptions under strong probabilistic constraints. In particular, when the latent logits for certain diseases are excessively large, the model not only suppresses false positives but also amplifies existing **true positives**. As illustrated by the bar chart in Figure 2, this behaviour leads to verbose generations, which compromise the fluency and naturalness of the radiology report style. To strike a balance between clinical accuracy and lexical quality, we adopt $\beta = 0.5$ as the default setting.

Table 10: **Ablation study of the $\gamma$ hyperparameter.** $\alpha = 0.5$ and $\beta = 0.5$ are used as default values. **Best** and second-best results are bolded and underlined, respectively. $\gamma \in \{2, 5, 10, \texttt{null}\}$.

| $\gamma$ | Lexical Metric | | | Clinical Metric | | | | | |
|---|---|---|---|---|---|---|---|---|---|
| | ROUGE-L | BLEU | BERTScore | RadGraph-F1 | Temporal-F1 | RaTEScore | RadEval-BERT | CheXbert$_{F1}^5$ | CheXbert$_{F1}^{14}$ |
| 2 | 20.70 | 1.98 | 51.65 | 18.85 | 17.52 | 53.32 | 17.56 | 22.20 | 12.45 |
| 5 | **20.71** | 2.05 | **51.67** | 18.98 | **17.65** | **53.35** | 17.55 | 25.52 | 14.52 |
| 10 | 20.70 | **2.10** | 51.62 | **19.01** | 17.58 | 53.32 | 17.50 | **27.05** | 16.02 |
| null | 20.35 | 2.06 | 51.40 | 18.85 | 17.41 | 53.14 | **17.60** | 26.21 | **16.35** |

**Effect of $\gamma$ on Guidance Strength**    As shown in Table 10, we evaluate the effect of varying $\gamma$, which controls the strength of the Diagnostic Plausibility Constraint in the second stage of Expert-informed Contrastive Decoding. We experiment with values of $\gamma \in \{2, 5, 10\}$ and also include a baseline where the constraint is removed entirely (denoted as $\texttt{null}$). As $\gamma$ increases, the plausibility threshold becomes more relaxed, allowing the model to be more influenced by the expert model's confidence scores. This, in turn, amplifies the suppression of false positives and the reinforcement of true positives, particularly in borderline cases. While some metrics such as RadEval-BERT and CheXbert$_{F1}^{14}$ peak at lower constraint strengths, the overall performance in both lexical and clinical metrics is best balanced when $\gamma = 10$. Therefore, we adopt $\gamma = 10$ as the default setting, corresponding to a clinically meaningful threshold for severe diagnostic evidence.

### E.2 ROBUSTNESS TEST OF CLINICAL CONTRASTIVE DECODING WITH RANDOM PRIOR

Since our method relies on guidance signals from a task-specific expert model, and Section 5.3 has demonstrated that stronger experts contribute to improved MLLM performance, it is important to assess how CCD behaves when this guidance becomes unreliable. To this end, we conduct an adversarial ablation study, where the expert model is deliberately degraded by replacing its outputs with randomly generated signals. This setting allows us to evaluate the robustness of CCD under faulty or misleading expert supervision. This experiment is conducted using MAIRA-2 (Bannur et al., 2024) as the backbone model, evaluated on the MIMIC-CXR (Johnson et al., 2019b) dataset for the radiology report generation task, with CCD hyperparameters kept at the default values.

Table 11: **Adversarial ablation study of CCD.** The *Random Setting* indicates that the signals from the expert model are replaced with randomly generated values. **Best** and second-best results are bolded and underlined, respectively.

| Method | Lexical Metric | | | Clinical Metric | | | | | |
|---|---|---|---|---|---|---|---|---|---|
| | ROUGE-L | BLEU | BERTScore | RadGraph-F1 | Temporal-F1 | RaTEScore | RadEval-BERT | CheXbert$_{F1}^{5}$ | CheXbert$_{F1}^{14}$ |
| **Baseline** | 19.57 | 1.61 | 49.56 | 16.23 | 12.11 | 50.82 | 16.96 | 16.14 | 10.57 |
| + *Random Setting* | 20.04 | 1.39 | 51.57 | 16.51 | 14.07 | 50.29 | 16.85 | 16.46 | 10.29 |
| + CCD | **20.70** | **2.10** | **51.62** | **19.01** | **17.58** | **53.32** | **17.50** | **27.05** | **16.02** |

As shown in Table 11, although the random setting introduces mild fluctuations in performance, there is no significant degradation across lexical or clinical metrics. This demonstrates that CCD does not substantially impair the MLLM's generation quality, even when the expert signal is adversarial. These findings highlight the robustness and compatibility of our method: **it enhances downstream performance only when the expert provides meaningful guidance, while gracefully falling back to the base model's behaviour otherwise.**

## F BALANCING ACCURACY AND AMBIGUITY

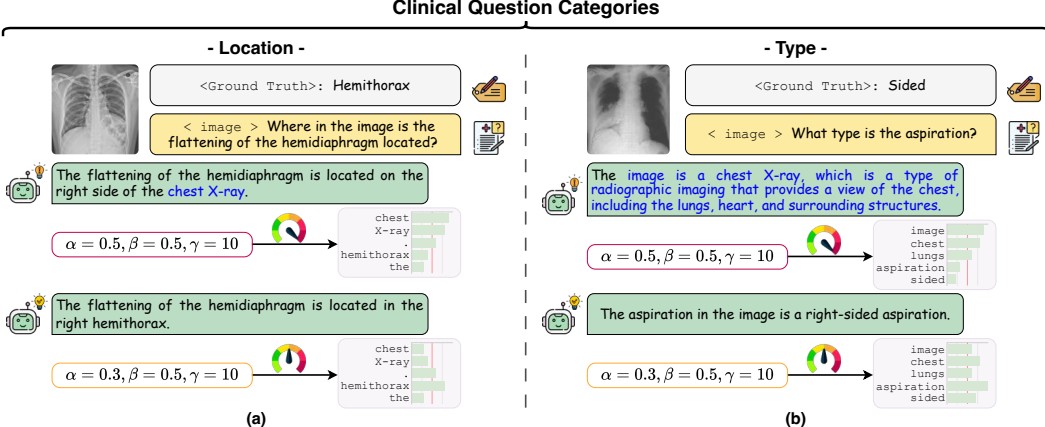

Figure 4: Illustration of additional VQA cases with **CCD**, using LLaVA-Med (Li et al., 2023a) as the baseline. **(a)** is a *location*-specific question and **(b)** a *type*-specific question. $\alpha$, $\beta$, and $\lambda$ denote CCD hyperparameters during inference. Model outputs that are vague or under-specified (i.e., partially correct but lacking clinical precision) are highlighted in blue. Latent logit ratio plots illustrate token-level differences, with **(a)** highlighting the final term and **(b)** the second token. In both cases, the top-5 overlapping tokens across two hyperparameter settings are shown as examples. The chest X-ray is blurred to preserve privacy and minimise visual discomfort.

In this section, we provide additional analysis of the two question categories that exhibited slight performance drops. As shown in Table 3, although CCD improves the overall performance of LLaVA-Med (Li et al., 2023a) on the Medical-CXR-VQA benchmark (Hu et al., 2024), two of the six evaluated categories, namely *Location* and *Type*, show a marginal decrease in accuracy. As mentioned in Section 5.1, we adopt a fixed set of hyperparameters across all models and tasks to ensure a fair comparison. As further discussed in Appendix D.2, we deliberately avoid tailoring hyperparameters to individual models or question types. While this promotes generality and ease of use, it may also limit performance in specific question categories that are more sensitive to decoding configurations. This trade-off reflects our focus on plug-and-play over task-specific tuning.

Figure 4 presents representative examples from the two categories with degraded performance under the default CCD hyperparameter setting. Some answers, although marked as incorrect, contain ambiguous yet clinically reasonable descriptions (highlighted in blue). While technically incorrect under strict evaluation criteria, these responses are not clearly erroneous but instead reflect overly cautious or broadly phrased interpretations, leading to borderline misjudgements.

Upon examining the latent logits distribution[5], we observe that ground-truth tokens often have lower activation scores compared to tokens associated with more generic symptom labels. This behaviour arises from the initial anchor stage of CCD, which introduces a strong bias toward common CheXpert-related symptoms, resulting in conservative outputs. In this case, the model tends to favour frequently seen "true positive" tokens and under-represents more specific or context-dependent concepts, leading to what can be considered "dummy" false negatives.

To explore this further, we reduce the control strength of the first decoding stage by adjusting $\alpha$ from 0.5 to 0.3. This softens the expert guidance, allowing the model to generate more accurate and specific answers in both *Location* and *Type* categories. These findings suggest that different question types may exhibit varying levels of sensitivity to CCD's control parameters.

While fine-grained control can improve performance for specific question categories, it also underscores a broader challenge: achieving the right balance between conservative and expressive generation. Overly cautious answers may avoid clinical errors but sacrifice specificity, while assertive responses can introduce misleading or incorrect information. This trade-off leads to an important question in the context of medical AI: **What constitutes a "better" response in radiology MLLMs?**

> *"It's better to be roughly right than precisely wrong."*
>
> — *Carveth Read*
> *Logic: Deductive and Inductive*

This quote from Read (1914) aptly reflects the philosophy behind our decoding strategy. In high-stakes settings such as radiology, generating responses that are somewhat ambiguous but clinically plausible is often preferable to confidently asserting inaccurate conclusions. From a system-level perspective, this approach improves overall reliability without compromising safety. **CCD** navigates this space by providing a balanced mechanism that moderates the influence of expert signals during generation while maintaining flexibility. Ultimately, this reflects a broader tension in aligning AI behaviour with clinical reasoning, where ambiguity, uncertainty, and contextual judgment are fundamental to the decision-making process.

## G    EXTENDED DISCUSSION ON LIMITATIONS

While our study demonstrates promising results across multiple benchmarks, several limitations merit consideration, particularly in clinical applications where the requirements for safety, reliability, and interpretability are significantly more stringent than in general-purpose AI tasks.

First, both the MIMIC-CXR (Johnson et al., 2019b) and Medical-CXR-VQA (Hu et al., 2024) datasets originate from the same institution, the Beth Israel Deaconess Medical Center. This may introduce institution-specific biases in patient demographics, imaging protocols, and clinical reporting practices, potentially limiting the generalisability of our findings to other healthcare settings with differing patient populations or workflows. Our choice of these datasets is primarily motivated by their unique status as the only publicly available sources that comprehensively align chest X-ray images with detailed free-text reports and structured question-answer annotations.

Second, all evaluations in this study rely on automatic metrics that serve only as relative references to the ground truth. While this approach is consistent with existing literature on radiology-focused MLLM evaluation, more robust validation would benefit from reader studies or expert review by licensed radiologists to further assess the clinical plausibility and safety of the generated outputs.

Third, our experiments rely on publicly available models such as MAIRA-2 (Bannur et al., 2024), of which only the 7B variant is currently open-sourced. Larger versions (e.g., MAIRA-2 13B) are not yet publicly accessible. Meanwhile, many high-performing models are only accessible via third-party APIs, which limits our ability to perform controlled experiments and to investigate scaling behaviours within our framework. This is particularly restrictive for our method, which requires direct access to the model's latent logits space in order to apply targeted modifications. Furthermore,

---

[5]This differs from the logit plots in Figure 2, where the truncation point is defined as the token immediately following the model's first output of a symptom phrase, namely after "Yes, the chest X-ray image shows ...".

since our evaluations are conducted in a shared offline environment, online latency in real-world deployments may differ significantly.

Moreover, while CCD demonstrates strong performance with empirically chosen hyperparameters, it currently lacks an adaptive mechanism to adjust control strength based on task complexity, prompt context, or model uncertainty. Exploring dynamic control strategies that can respond to such internal or external signals may be a promising direction for future work—particularly for achieving a better trade-off between clinical accuracy and generation fluency across diverse applications.

In addition, most radiology MLLMs and expert models are trained on well-curated datasets like MIMIC-CXR (Johnson et al., 2019b), where image quality is standardised and acquisition conditions are controlled. As noted in Appendix A.2, these models do not cover other modalities such as computed tomography (CT), magnetic resonance imaging (MRI), or ultrasound. However, real-world clinical practice often involves lower-quality inputs, including portable X-rays or images from heterogeneous equipment. Evaluating robustness under such distribution shifts remains an important direction for future research.

In conclusion, this work takes a step toward advancing radiology-oriented multimodal language models (MLLMs) toward physician-level reasoning. Our results show that even current state-of-the-art models can be further improved by incorporating domain-specific expert models, as demonstrated by our proposed **CCD** framework. Although generative foundation models are developing rapidly, we believe that specialised expert models are still a necessary part of medical AI, especially in safety-critical tasks like medical imaging. This study presents a possible way to combine the strengths of both types of models to improve clinical accuracy.

## H  ADDITIONAL STATEMENT: THE USE OF LARGE LANGUAGE MODELS (LLMs)

In this work, we used a generative AI model to assist with colour editing and refinement of the icons in Figure 1, 2, and 4. This step was intended solely to improve visual clarity and enhance the overall readability of the figures. The use of this tool was strictly limited to visual presentation and did not influence the scientific content, analysis, or experimental results presented in the paper. We also employed Overleaf's AI assistant to ensure spelling and grammar consistency throughout the manuscript, using UK English conventions.

## I  ADDITIONAL STATEMENT: SPECIAL ACKNOWLEDGEMENTS

We would like to express our sincere gratitude to the reviewers on OpenReview[6] for their insightful comments and suggestions on this work. We strongly encourage readers to consult the public review discussions, which not only provide valuable context for understanding our contributions, but may also serve as a source of inspiration for future research.

---

[6]https://openreview.net/forum?id=eEnW7lUXxY

