# OpenReview forum: "CCD: Mitigating Hallucinations in Radiology MLLMs via Clinical Contrastive Decoding"
_ICLR.cc/2026/Conference — ICLR 2026 Conference Withdrawn Submission_

### Official Review · Reviewer_aSHU · 2025-10-29

**Soundness:** 2
**Presentation:** 2
**Contribution:** 3
**Rating:** 4
**Confidence:** 4

**Summary:**

This paper presents a contrastive decoding approach tailored to radiology report generation (RRG) called. clinical contrastive decoding (CCD). The CCD framework adjusts the token logits of a pretrained multimodal large language model (MLLM) using the outputs of a pretrained image classifier, enabling a training-free method to enhance the generations of the MLLM. The authors additionally conduct an analysis on a potential source of hallucination in the generations of the MLLM within the context of RRG.

**Strengths:**

●	 Interesting idea to understand the potential to utilize a pretrained classifier to enhance model generation without retraining the LLM. Interesting approach to modify the token logits backed by context provided evidence of a domain “expert” image model.

●	 Valuable analysis to try to quantify the impact of potentially misleading information in the model prompt. Empirical results do demonstrate differences, though perhaps still an open question as to why those differences exist.

●	 The manuscript is well prepared with generally clear writing, equations, figures, and tables.

**Weaknesses:**

●	The authors’ claim on the impracticality of retrieval doesn’t seem well founded, it’s  poorly cited and in reality, computing similarities over precomputed embeddings of images or text should be relatively cheap compared to the LLM inference that the system needs to do anyways.

●	While section 3 and table 1 provide interesting empirical evidence that the generations are sensitive to the prompts, the conclusions stated by the authors are more conjecture than evidence-based claims. For example, lines 216-217 claim clinical terminology and standardized phrasing is more prevalent in the history or technique sections, yet it is easy to observe similar standardized clinical phrasing in the other note sections exemplified in figure 1a. Additionally, in the paragraph on lines 222-231, the authors claim certain findings are more prevalent in different note sections. The authors fail to discuss the general increase in performance for cardiomegaly, consolidation, and edema when including all note sections in the prompt. Further the claims that are made by the authors, in terms of why atelectasis and pleural effusion do worse, should be quantified rather than relying on citing a review article from 1996 that only reviews atelectasis. It would seem that there is a lack of clinical background and context for the authors to make broad claims about clinical practice.

●	It is confusing how the CCD framework is using the expert model predicted probabilities in the ECD arm of the method. Equation 5 shows the clipped bias for a single $\tilde{l}_i$ being “uniformly added to the token logits”; how is the symptom label “selected” (line 337)? Does the label change depending on the step t? Figure 2 seems to present the use of all labels for every step t, so it seems further clarification on this fundamental component of the method is warranted.

●	Relatedly, given the use of the 14 CheXpert labels as the grounding source of information, why should CCD mitigate hallucinations such as the example “AP and lateral” vs “single frontal” provided in Figure 1a or help the model determine the type of atelectasis in Figure 1b? It is additionally unclear based on the presented results and methodology whether the authors actually evaluated whether CCD could mitigate these hallucinations in the context of the varying prompt information. See related question below on table 2 vs table 1 results.

●	The presentation of relative change in performance overstates results and should be tempered. The absolute change in F1RadGraph is ~2.8%. Additionally, while this work aims to demonstrate the relative gain of CCD over its baseline model, it would still be beneficial to understand the magnitude of the gain relative to the more complex approaches the authors aim to beat. Some of these results are presented in Appendix Table 6, however these results demonstrate much weaker results for generalizability of CCD.

**Questions:**

●	In your experiments in section 3 on prompt-induced hallucination, was the type of clinical context provided in the prompt? I.e. was the note section header given as part of the clinical context? This would help the model better understand the information it was given, otherwise the results of section 3 may be confounded by the lack of contextualization.

●	Figure 2 seems to show the CheXpert labels are single tokens, is that the case? I would suspect not, given then the need to retrain the tokenizer and model, perhaps this should be clarified.

●	Lines 344-345 mention the use of different decoding components in the logits processor, yet line 378/9 state the use of greedy decoding. This should be clarified as the decoding strategies used in the logits processor are not the same as the final decoding approach, yet it still has an impact on the output logits.

●	It is unclear if the radiology report generation results presented in Table 2 includes any of the context prompts that were assayed in Table 1. It would be a stronger result to understand how CCD has mitigated those hallucinations in the presence of potentially counterfactual information.

---

> ### Author Response · Authors · 2025-11-19
> **Individual Response to Reviewer aSHU - Thank you!**
>
> Thank you for your valuable suggestion and for your service in reviewing our paper. Our detailed responses to your comments are provided as follows:
>
> ***
>
> # W1: Retrieval
>
> Thank you for raising this point.
>
> > ***"The authors' claim on the impracticality of retrieval doesn't seem well founded ... should be relatively cheap compared to the LLM inference that the system needs to do anyways."***
>
> We would also like to clarify that the **additional cost referred to in our response is not related to computational efficiency**. As stated in Line 125, "...strict privacy regulations exacerbate the scarcity and imbalance of clinical training data..." and Line 141/142 further notes that such approaches "...raise privacy or security concerns..." and "...in out-of-distribution settings or when adapting to new benchmarks." Recent concurrent work [1] has also expressed similar views. Therefore, **the motivation of this work is not centred on reducing computational cost.**
>
> Moreover, Line 985 explicitly states: "Our focus is on test-time decoding. Therefore, we do not compare with ... or retrieval-based augmentation..." Consequently, discussions involving non-testing-time methods, such as additional data collection, external retrieval, or privacy-sensitive pipelines, lie outside the scope of this study. Even if retrieval-based methods are computationally inexpensive, they nevertheless fall outside the model decoding strategy that our work focuses on.
>
> [1] Kim et al., "Rethinking Retrieval-Augmented Generation for Medicine: A Large-Scale, Systematic Expert Evaluation and Practical Insights" 2025-Nov
>
> ***
>
> # W2: Clinical Background
>
> Thank you for this meaningful suggestion.
>
> > ***"While section 3 and table 1 provide interesting empirical evidence that the generations are sensitive to the prompts, the conclusions stated by the authors are more conjecture than evidence-based claims."***
>
> We acknowledge that **our conclusions are based primarily on quantitative experiments** and do not include qualitative analyses such as radiologist assessments. As stated in Section 3 (Line 161), "we conduct empirical analyses to...prompt-induced medical hallucinations," and Section 3 is designed to validate this specific category of hallucinations. Our analysis is further scoped to the RRG task, as noted in Line 182: "we focus on the chest X-ray modality and the RRG task."
>
> As noted in Line 201, "We follow prior work and adopt a set of lexical and radiology-specific metrics..." Consistent with previous studies, this represents **the standard empirical evaluation practice in this domain**. Methods such as LLaVA-Rad [2] and RADAR [3], as well as recent evaluations from Google [4], also rely on report-level scores for assessing radiology report generation task.
>
> [2] Zambrano Chaves et al., "A clinically accessible small multimodal radiology model and evaluation metric for chest X-ray findings" Nature Communications 2025
>
> [3] Hou et al., "RADAR: Enhancing Radiology Report Generation with Supplementary Knowledge Injection" ACL 2025
>
> [4] Google, "MedGemma Technical Report" 2025
>
> > ***"It would seem that there is a lack of clinical background and context for the authors to make broad claims about clinical practice."***
>
> We acknowledge that **we are not clinically trained experts, which further underscores the importance of professional radiologist assessment**. In Appendix G (Line 1477: "...more robust validation would benefit from reader studies..."), we explicitly recognise this limitation and note that human evaluation would indeed strengthen the validation. However, **such studies are prohibitively expensive, and we are currently unable to support them.** We intend to incorporate expert reader studies in future work to further evaluate our method, and we will actively seek the necessary funding and opportunities to enable such assessments.
>
> ***
>
> # W3: CCD Framework
>
> We would like to clarify this point further.
>
> > ***"...how is the symptom label 'selected' (line 337)?"***
>
> **We select symptom labels directly from the expert model's default output.** As mentioned in Line 292, "These expert-provided symptom labels are filtered using a default threshold (e.g., $s_i > 0.5$), and the selected labels are...". This threshold corresponds to the recommended configuration used by the underlying symptom classifier.

---

> > ### Comment · Reviewer_aSHU · 2025-11-26
> >
> > Thank you for your responses to our comments and for your detailed responses to the other reviewers.   After reviewing, I'm going to maintain my original ratings.

---

> > > ### Author Response · Authors · 2025-11-27
> > >
> > > Dear Reviewer aSHU,
> > >
> > > Thank you very much for your follow-up. We sincerely appreciate your time and service to the community in reviewing our paper. If there are any remaining concerns that we may have overlooked, we would be happy to provide further clarification.
> > >
> > > Best regards,
> > >
> > > All Authors

---

> ### Author Response · Authors · 2025-11-19
>
> ## (following above)
>
> > ***"Does the label change depending on the step `t`?"***
>
> **The labels do not change with the decoding step `t`**. As noted in Section 4.1 (Line 289), "we use a DenseNet-based classifier ... given v " where `v` is the image. Line 291 further states that "from this expert model, we extract a set of clinical labels...". Therefore, **for a single chest X-ray, the labels obtained from the expert model are fixed from the outset**.
>
> In contrast, `t` is introduced only in the context of language model decoding. As described in Line 269 ("Preliminaries of MLLM Generation") and Line 277 ("..the output logits at decoding step t are denoted as...”), the variable `t` refers exclusively to the autoregressive decoding steps of the LLM, not to the expert classifier.
>
> > ***"Figure 2 seems to present the use of all labels for every step `t`,..."***
>
> **Yes**, the initial expert-provided labels influence all subsequent decoding steps `t`. As illustrated in Figure 2, because the MLLM is an autoregressive model that generates text token by token, we must apply the guidance at every decoding step. This is why the notation `t` is introduced only after the model begins producing output tokens. **The left side of Figure 2 marks the "log-softmax" operation**, which indicates that `t `governs the propagation of logits through each decoding step. Representing the process in this way is therefore a more accurate and consistent formulation.
>
> ***
>
> # W4: Mitigate Hallucination
>
> We would like to clarify this point further.
>
> > ***"Why should CCD mitigate hallucinations such as the example 'AP and lateral' vs 'single frontal' provided in Figure 1a "***
>
> We **do not use these clinical sections** (e.g., Figure 1a, Line 58: "Portable anteroposterior (AP) and lateral chest radiograph") **as part of the model input**, thereby avoiding the ambiguity they introduce. Without these misleading sections, the model is able to produce the correct output such as 'single frontal'.
>
> As noted in Section 3, Line 85, "prompt-induced medical hallucinations, triggered by clinically implausible or ambiguous prompts..", arise from the clinical sections contained in the dataset itself. Examples such as 'AP and lateral' originate from these raw report sections and can mislead the model at input. **The most direct way to eliminate such prompt-induced hallucinations is therefore simply not to use these clinical sections as part of the model input.** Accordingly, as illustrated in Figure 1a, our default prompt is used without any clinical context. **The only message the model receives is**: "Provide a detailed description of the findings in the radiology image."
>
> > ***"why help the model determine the type of atelectasis in Figure 1b?"***
>
> As noted in Line 291, "the selected labels are then used to construct a concise anchor prompt (e.g., ‘Attention to the following clinical instructions: Atelectasis, Cardiomegaly...’)." Therefore, if a chest X-ray contains a finding such as atelectasis, the expert model could provide a corresponding signal.
>
> **The example in Figure 1(b) illustrates this effect:** `"What type is the atelectasis? Attention to the following clinical instructions: Atelectasis."` In addition to this anchor prompt, the second-stage probability signals further reinforce the guidance (Section 4.2). Together, these signals help the model focus on the relevant pathology and support its downstream reasoning.
>
> > ***"See related question below on table 2 vs table 1 results."***
>
> ***Please refer to the response to Question 4 below.***
>
> ***
>
> # W5: Performance
>
> Thank you for this constructive suggestion.
>
> > ***"The presentation of relative change in performance overstates results and should be tempered. The absolute change in F1RadGraph is ~2.8%."***
>
> We fully agree with your observation. Accordingly, **both the Abstract and Table 2 have been updated to report absolute rather than relative improvements.**
>
> > ***"It would still be beneficial to understand the magnitude of the gain relative to the more complex approaches the authors aim to beat."***
>
> We agree with this point. The strategies **we compare against are methods developed for the general domain**, and as shown in our experiments, several of them even reduce baseline performance when applied to radiology MLLMs. To the best of our knowledge, **at the time of submission, no prior work had proposed decoding strategies specifically designed to optimise radiology MLLMs**, and therefore no domain-specialised baselines were available for comparison.
>
> As noted in Line 154, "While effective in such domains, these methods struggle to mitigate medical hallucinations in radiology..." Furthermore, Appendix D.1 (Line 1157 "The following is a brief overview of these methods.") provides an overview and discussion of each competing method, highlighting **why general-domain decoding approaches are not directly effective in this clinical setting.**

---

> ### Author Response · Authors · 2025-11-19
>
> ## (following above)
>
> > ***"Some of these results are presented in Appendix Table 6, however these results demonstrate much weaker results for generalizability of CCD."***
>
> We acknowledge that **CCD exhibits some limitations in generalisation**. This arises because **the default hyperparameters used for CCD were selected based on empirical optimisation** for MAIRA-2 on the RRG task, and therefore may not be optimal for other models.
>
> As stated in Section 5.3 (Line 471), "the best empirical RadGraph-F1 score is achieved when both guidance strengths reach 0.5," and the ablation studies in Appendix E (Tables 7, 8, and 9) support this choice. Based on these observations, we adopt $\alpha = 0.5$ and $\beta = 0.5$ as the hyperparameter configuration for MAIRA-2. **Selecting hyperparameters through empirical validation is also a common practice for decoding strategies in the general domain**, and our choice aligns with prior work, such as VTI [5], in their hyperparameter design.
>
> In addition, we have already acknowledged this limitation in generalisation, consistent with the statement in Line 1280: "This suggests that choosing appropriate hyperparameters for each model is critical to achieving a balanced trade-off between lexical and clinical performance." **We have included further discussion of this point in Appendix G.**
>
> [5] Liu et al., "Reducing Hallucinations in Vision-Language Models via Latent Space Steering" ICLR Spotlight 2025
>
> ***
>
> # Q1: Experiments
>
> > ***"In your experiments in section 3 on prompt-induced hallucination, was the type of clinical context provided in the prompt? "***
>
> Yes, we have ensured that the experimental design remains rigorous, and **we will include all original information accordingly**. As illustrated in Figure 1, Line 62 specifies the default template: "Provide a detailed description of the findings in the radiology image. Following clinical context: <clinical section>" When a clinical section is present, the prompt includes all content shown in the top-right of Figure 1, as indicated in Line 54 ("Clinical Section"). Thus, the full prompt in such cases becomes:
> `"Provide a detailed description of the findings in the radiology image. Following clinical context: INDICATION:...TECHNIQUE:...COMPARISON:... HISTORY:..."`
>
> This accurately reflects the raw clinical sections used in the dataset and ensures that our experimental setup faithfully represents the original input conditions.
>
> ***
>
> # Q2: CheXpert Label
>
> > ***"Figure 2 seems to show the CheXpert labels are single tokens, is that the case? I would suspect not..."***
>
> Yes, you are correct — **they are simply abbreviations**. The abbreviations shown in the lower-left of Figure 2 are merely shorthand **name of the symptoms**, used for visual clarity and to avoid repeated long text in the figure. As stated in Line 291, "each $\ell_i$ denotes a finding (e.g., ‘Atelectasis’)", and Line 293 further clarifies that the anchor prompt is constructed as "Attention to the following clinical instructions: Atelectasis, Cardiomegaly...". Since these are standard free-text terms, the **LLM's tokenizer can recognise them without requiring any retraining.**
>
> #  Q3: Logits Processor
>
> Thank you for this meaningful question.
>
> > ***"Lines 344-345 mention the use of different decoding components in the logits processor, yet line 378/9 state the use of greedy decoding."***
>
> You are correct. **All our experiments employ greedy decoding**. We control this behaviour through the `LogitsProcessor()`, which governs decoding behaviour, sequence length, and related settings — **including greedy decoding**. This module is therefore an essential part of the generation framework and naturally incorporates the decoding strategy used in our experiments. **We have reiterated this point at Line 344 in the revised manuscript to improve clarity.**
>
> ***
>
> # Q4: Context Prompt
>
> Thank you for raising this question.
>
> > ***"It is unclear if the radiology report generation results presented in Table 2 includes any of the context prompts that were assayed in Table 1."***
>
> **For Table 1**, as stated in Line 198, "We adopt the default prompt shown in Figure 1(a) and use greedy decoding, the standard setting for radiology MLLMs. In each case, we append a different clinical section." Accordingly, **Table 1** compares the **default prompt with versions** that include the corresponding **clinical sections.** (e.g., `"Provide a detailed description of the findings in the radiology image. Following clinical context: INDICATION:...TECHNIQUE:...COMPARISON:... HISTORY:..."`)
>
> **For Table 2**, Line 273 states: "For the default prompt, we represent it as x...". Thus, **Table 2** evaluates the model using **only the default prompt** and does **not** include any additional clinical sections. (e.g., `"Provide a detailed description of the findings in the radiology image."`)

---

> ### Author Response · Authors · 2025-11-19
>
> ## (following above)
>
> > ***"It would be a stronger result to understand how CCD has mitigated those hallucinations in the presence of potentially counterfactual information."***
>
> In Line 85, we note that "prompt-induced medical hallucinations, triggered by clinically implausible or ambiguous prompts", **arise from the clinical sections contained in the dataset itself**. As described in Appendix B.1 (Line 1027), "Each report is preprocessed to extract five clinically relevant sections: Findings, Indication, Technique, Comparison, and History. This is done using pattern-matching heuristics based on the official preprocessing scripts."
>
> Given that these sections may contain incomplete or misleading information, **the most effective way to avoid such hallucinations is simply not to use these clinical sections as part of the prompt**. Instead, we rely on an expert model to produce clinically meaningful guidance signals from chest X-rays, including the “anchor prompt” described in Line 293. In short, **CCD does not depend on noisy initial clinical context.** Rather, it improves the MLLM's output quality by incorporating auxiliary signals that are controlled, as produced by the expert model.
>
>
> ***
>
> We sincerely hope that our responses have addressed your concerns. If we have left any notable points of concern overlooked, we would greatly appreciate your further feedback, and we will attend to these points. Additionally, we will incorporate all the clarifications and improvements mentioned above in the final manuscript. Thanks again for your thoughtful review and consideration.

---

### Official Review · Reviewer_H2Rc · 2025-11-01

**Soundness:** 4
**Presentation:** 3
**Contribution:** 2
**Rating:** 4
**Confidence:** 4

**Summary:**

The paper proposes Clinical Contrastive Decoding (CCD), a training-free, retrieval-free inference method to reduce medical hallucinations in radiology MLLMs. CCD plugs in a task-specific expert classifier (e.g., CheXpert-style DenseNet) and uses its predicted labels and probabilities to (1) inject a “clinical anchor” into the prompt (SCD) and (2) bias token logits with calibrated logit offsets (ECD). Evaluated on MIMIC-CXR, IU-Xray, and CheXpert Plus across RRG and CXR-VQA, CCD improves several clinical metrics (e.g., up to +17% RadGraph-F1 on MIMIC-CXR) without retraining the base MLLM.

**Strengths:**

1. The paper is well-motivated. Instead of just proposing a new method, the authors first dedicate Section 3 to diagnosing a specific, tangible problem: MLLMs "overreact to clinical context" , and the clinical sections in reports "are not always reliable sources of guidance". This empirical analysis provides a strong justification for their solution.
2. The proposed CCD framework is both training-free and retrieval-free. This is a significant advantage in the medical domain, as it avoids the cost, privacy concerns, and complexity associated with retraining models or maintaining large retrieval databases.

**Weaknesses:**

1. The empirical motivation in Table 1, while interesting, is not as compelling as it could be. The authors demonstrate that adding noisy or counterfactual context (like "comparison" or "technique" sections) degrades performance. This is somewhat expected, as the model is not given the prior images needed for comparison, confusing its generation task.
A stronger motivation for CCD would be to complete this analysis by showing the opposite:
First, demonstrate that feeding the model clean, structured labels (like those from an expert classifier) as prompt context actually improves performance over the baseline in Table 1.
Then, contrast this with the performance degradation when using the noisy, original report sections.

This would more forcefully prove the central hypothesis: MLLMs are harmed by bad context but benefit from good context, thereby providing a stronger justification for the CCD framework.

2. The framework's performance is fundamentally coupled to the quality and design of the expert model, which introduces several limitations:
a. The framework's effectiveness is capped by the expert model's accuracy. As the authors' own ablations suggest, a stronger expert yields better results. Conversely, a weak or systematically biased expert model could, in principle, steer the MLLM to replicate its own errors, potentially propagating biases.

b.  The use of a simple > 0.5 probability threshold to select expert labels for the anchor prompt is suboptimal, especially for highly imbalanced medical datasets where clinically important findings are often rare. More sophisticated, data-driven thresholding techniques (e.g., maximizing F1-score) would be better suited to calibrate the expert guidance.
[1] The limits of fair medical imaging AI in real-world generalization. Nature Medicine.
[2] Algorithmic encoding of protected characteristics in chest X-ray disease detection models. Lancet

c. The CheXpert ontology’s 14 pathologies provide limited morphological and positional detail (e.g., scarring, lesion location), which explains the observed performance dips in VQA categories like Location and Type.
This reflects an inherent ceiling for CCD’s guidance. The authors could extend their framework by incorporating expert findings for anatomical regions and observation types (using RADGRAPH), inspired by:
[1] Anatomy-Guided Weakly-Supervised Abnormality Localization in Chest X-rays. MICCAI 2022
[2] Dividing and Conquering a BlackBox to a Mixture of Interpretable Models: Route, Interpret, Repeat. ICML 2023
[3] Distilling blackbox to interpretable models for efficient transfer learning. MICCAI 2023

3. The evaluation could be expanded to include LLM-as-a-Judge metrics, particularly the recently proposed GREEN framework (EMNLP 2024, “Generative Radiology Report Evaluation and Error Notation”). Such metrics capture clinically nuanced correctness and could provide complementary insights beyond traditional lexical or structured metrics.

4. The paper positions itself as a solution to prompt-induced hallucinations, as identified in Table 1 where "comparison" sections cause errors. However, it is unclear how the proposed CCD framework—which guides the MLLM based on findings in the current image—would explicitly prevent the model from spontaneously hallucinating references to past exams. This common and problematic type of hallucination seems unaddressed by the current mechanism.


5. The paper currently lacks consistent figure references and variable definitions.
Figure 2 is cited as having panels (a) and (b), but no such markings are visible.
The variable c in line 298 is undefined; it appears to refer to the structured labels from the expert model, which should be explicitly stated.
Improving figure annotations and notational clarity will make the methodology easier to follow.

**Questions:**

1. Since the framework’s success depends heavily on the expert classifier, how sensitive is CCD to expert calibration and label-imbalance effects?

2. Given that the 14 CheXpert labels lack morphological or positional coverage, do the authors plan to incorporate anatomical and observation-level findings (e.g., from RADGRAPH or anatomy-guided classifiers) as additional expert concepts?
How might CCD generalize to these richer ontologies, which could address the VQA “Location/Type” performance dips?

3. Have the authors considered evaluating with LLM-as-a-Judge metrics such as GREEN (EMNLP 2024)?

---

> ### Author Response · Authors · 2025-11-19
> **Individual Response to Reviewer H2Rc - Thank you!**
>
> Thank you for your valuable expertise in this space and your service in reviewing our paper! Our detailed response to your concerns/questions is listed as follows:
>
> ***
>
> # W1: Motivation
>
> We acknowledge that this line of reasoning is indeed helpful for strengthening the paper.
>
> > ***"A stronger motivation for CCD would be to complete this analysis by showing the opposite: First, demonstrate that feeding the model clean, structured labels (like those from an expert classifier) as prompt context actually improves performance over the baseline in Table 1. Then, contrast this with the performance degradation when using the noisy, original report sections."***
>
> Our current motivation is grounded in the observation introduced in Line 85, where we state that "...prompt-induced medical hallucinations, triggered by clinically implausible or ambiguous prompts..." These sections originate from the original reports, which motivates the ablation in Table 1 in Section 3.
>
> **Regarding your suggestion to feed clean, structured labels from an expert classifier,** if we understand correctly, this refers to providing these labels directly to the model without applying any hyperparameter control.  **Within our framework, this corresponds to the special case of setting $\alpha = 1$ (instead of the default 0.5),** meaning that the model relies entirely on the expert-provided labels. In Appendix D.2, Table 7, when the MLLM fully depends on these expert labels ($\alpha = 1$,), performance improves on certain metrics. However, with the default setting of $\alpha = 0.5$, a broader range of clinically meaningful metrics, including RadGraph-F1, reaches its peak performance.
>
>
> # W2: Performance Limitations
>
> Thank you for your detailed and insightful comments.
>
> > ***"a. The framework's effectiveness is capped by the expert model's accuracy ... Conversely, a weak or systematically biased expert model could, in principle, steer the MLLM to replicate its own errors, potentially propagating biases."***
>
> We did take this into consideration and **conducted robustness tests to evaluate the effect of unreliable expert signals on the MLLM**. As shown in Appendix E.2, Line 1387 reports that "although the random setting introduces mild fluctuations in performance, there is no significant degradation across lexical or clinical metrics," consistent with the results in Table 10 (where 6 metrics increase and 4 show slight decreases compared with the baseline). Line 1390 further notes that "it enhances downstream performance only when the expert provides meaningful guidance, while gracefully falling back to the base model's behaviour otherwise."
>
>
> > ***"b: The use of a simple > 0.5 probability threshold to select expert labels for the anchor prompt is suboptimal..."***
>
> We fully agree with your point, and we have taken this aspect into account. The "threshold" here refers to the final probability outputs of the expert model, **which already incorporate threshold-based normalisation**. As noted in Line 288, the DenseNet-based classifier applies symptom-specific thresholds and sigmoid normalisation during inference. These thresholds are implemented directly in the official TorchXRayVision [1] model, **where each symptom category is associated with its own calibrated threshold in the classifier's forward pass** (see the [official implementation line 343-357](https://github.com/mlmed/torchxrayvision/blob/df3c84c8c31bebbeb8696d400c862e615110bdfc/torchxrayvision/models.py#L343C1-L357C1) for details).
>
> [1] Cohen et al, "TorchXRayVision: A library of chest X-ray datasets and models" 2021

---

> ### Author Response · Authors · 2025-11-19
> **following above**
>
> > ***"c: The CheXpert ontology's 14 pathologies provide limited morphological and positional detail, which explains the observed performance dips in VQA categories like Location and Type. The authors could extend their framework by incorporating expert findings for anatomical regions and observation types (using RADGRAPH)"***
>
> Thank you for this highly constructive suggestion. Our contribution primarily aims to show that **incorporating an expert model can enhance MLLM performance on downstream tasks**, rather than covering a broader set of label categories. For this study, we therefore use the 14-label CheXpert symptom-level classifier. **This choice remains flexible, and the module can be adapted to future expert models should more comprehensive label sets become available.** As stated in Line 287, "we focus on the 14 pathology labels defined in the CheXpert ontology..." Symptom-level signals are the only form of guidance that the expert model can provide within the scope of this study.
>
> **Regarding the slight drop in Location/Type about VQA**, we also highlight this in the caption of
> Table 3.  In Appendix F, Figure 4 (Line 1421) further explains that in these examples, "model outputs that are vague or under-specified (i.e., partially correct but lacking clinical precision)" lead to a dip that is largely an artefact of the evaluation procedure rather than a genuine performance drop. Additionally, Appendix A provides a detailed discussion of our research aims (A.1) and research scope (A.2). As stated in Line 987, "Nor do we attempt to eliminate all forms of medical hallucination." Incorporating anatomical-region or observation-type signals therefore falls outside the scope of the present study.
>
> **We agree that incorporating richer structural cues, such as RadGraph-based information, is an excellent direction.** This aligns well with our planned future work, and we would welcome the opportunity to collaborate in exploring this direction further if possible.
>
> ***
>
> # W3: Evaluation
>
> Thank you for raising this point.
>
> > ***"The evaluation could be expanded to include LLM-as-a-Judge metrics, particularly the recently proposed GREEN framework..."***
>
> We agree and have added a GREEN score to assess the improvement over the baseline. Due to page limitations, this analysis has been included in Appendix D.1.
>
>
> |               | GREEN |
> |----|----|
> | MAIRA-2   | 18.03 |
> | + CCD     | 19.14 |
>
> For the original manuscript, Section 3 (Line 202) states that "We follow prior work and adopt a set of lexical and radiology-specific metrics...", which are the standard and cost-effective evaluation practices in this domain. **Our setup follows prior work such as MAIRA-2 [2], RADAR [3], and LLaVA-Rad [4], using the overlapping lexical and clinical metrics adopted in these studies to ensure broad and consistent coverage.**
>
> Using more compute-intensive evaluation methods (e.g., GREEN or other LLM-based evaluators such as GPT-5) typically requires additional GPU resources or API costs. As mentioned in Line 1085, our experiments were conducted under limited computational capacity and inference-time constraints, using only two low-memory GPUs for all MLLM evaluation. **Requesting further GPU resources or significantly extending the computation time is currently beyond our affordable budget.**
>
> [2] Bannur et al., "MAIRA-2: Grounded Radiology Report Generation" 2024
>
> [3] Hou et al., "RADAR: Enhancing Radiology Report Generation with Supplementary Knowledge Injection" ACL 2025
>
> [4] Zambrano Chaves et al., "A clinically accessible small multimodal radiology model and evaluation metric for chest X-ray findings" Nature Communications 2025
>
> ***
>
> # W4: Framework
>
> We would like to clarify this point further.
>
> > ***"However, it is unclear how the proposed CCD framework—which guides the MLLM based on findings in the current image—would explicitly prevent the model from spontaneously hallucinating references to past exams."***
>
> Our method **avoids prompt-induced hallucinations at the model input** by preventing any downstream influence from the noisy or misleading clinical sections present in the original reports. In other words, it does not use this metadata.
>
> In Section 3 (Line 214, "Hallucination Drivers: Clinical Context Sensitivity"), we first demonstrate empirically that even when key clinical sections from the original reports (e.g., Comparison) are provided as additional context, the model's performance does not improve. Therefore, as stated in Line 197, "We adopt the default prompt shown in Figure 1(a)...", and Line 273 clarifies that "For the default prompt, we represent it as…". Under our setup, the model receives no clinical sections from the original reports during inference; **it only processes a single instruction, "Provide a detailed description of the findings in the radiology image."** This design ensures a clean and controlled evaluation setting, free from contamination by noisy clinical text.
>
> ***

---

> ### Author Response · Authors · 2025-11-19
> **following above**
>
> # W5: Figure & Variable Consistent
>
> Thank you very much for this suggestion. **In the revised version, we will increase the colour contrast in Figure 2 and emphasise the relevant description in the caption using bold text.**
>
> > ***"Figure 2 is cited as having panels (a) and (b), but no such markings are visible..."***
>
> In Figure 2, we have marked **(a)** and **(b)** above the corresponding model outputs on the right-hand side. Because the textual content is relatively long, we explicitly reference these annotations in the figure caption (Line 256).
>
> > ***"The variable c in ine 298 is undefined..."***
>
> We did define the variable `c` just above Line 298. Specifically, Lines 293–294 state: "construct a concise anchor prompt...'Atelectasis, Cardiomegaly, ...', denoted as `c`..." which clarifies the meaning of this notation.
>
> ***
>
> # Q1: Expert Sensitive
>
> > ***"...how sensitive is CCD to expert calibration and label-imbalance effects?"***
>
> Our results indicate that **CCD is not noticeably sensitive to label-imbalance effects**, as imbalanced expert signals do not introduce any notable performance degradation. As noted previously in our response to **W2-a,** the same reasoning applies here. In Line 485, we refer readers to Appendix E for detailed results, including the random-signal robustness test. Due to space limitations, these analyses are provided in the appendix. We have already discussed this issue thoroughly in Appendix E.2 (Robustness Test of Clinical Contrastive Decoding with Random Prior). In this experiment, we evaluate how the MLLM behaves when the expert signal becomes unreliable. In other words, **the expert model primarily provides auxiliary guidance and does not interfere with the MLLM's original capabilities**.
>
> ***
>
> # Q2: Additional Expert
>
> > ***"Do the authors plan to incorporate anatomical and observation-level findings as additional expert concepts? "***
>
> Indeed, **we plan to incorporate anatomical- and observation-level signals from additional expert models**, and this direction is highly relevant to our next line of work. We would welcome the opportunity to explore this together in the future. This constructive point is similar to the concern raised in **W2-c**. As already stated in Section 4.1 (Line 287), the expert model used in our study operates at the symptom level, which serves as the classifier to validate CCD in the present work.
>
> > ***"How might CCD generalize to these richer ontologies, which could address the VQA “Location/Type” performance dips?"***
>
> **We agree that richer ontologies have the potential to further improve performance on specific tasks.** We have already provided a detailed discussion of the "VQA Location/Type performance dips", as also noted in our response to **W2-c**.
>
> For this reason, developing or evaluating expert models specialised in morphology or topography **lies beyond the scope of the current study**. Due to space limitations, the main manuscript cannot elaborate further, but Appendix A already provides detailed descriptions of both our Research Aims and Research Scope. In Appendix A.1 (Line 984), we explain that "...mitigation of a subset of hallucinations can lead to meaningful gains in overall task performance..." and Line 991 notes that "...this is because most findings are concentrated in frontal-view chest X-rays, whereas lateral-view images provide less diagnostic signal for common conditions [2]."
>
> ***
>
> # Q3: LLM-as-a-Judge metrics
>
> > ***"...considered evaluating with LLM-as-a-Judge metrics such as GREEN"***
>
> **Yes, we will consider incorporating this evaluation protocol**, as also noted in our response to **W3**. The results have been added in the revised manuscript. However, resource constraints during this study made it difficult to request additional GPU hours, and API-based evaluators (e.g., RadFact [2], CheXPrompt [4]) would incur substantial additional cost.
>
> **Therefore, we adopted as many overlapping lexical and clinical metrics from prior work as possible,** including recent metrics such as RadEvalBERTScore [5]. With broader community adoption, we expect to integrate such evaluation protocols in future research. Nonetheless, using LLMs as evaluators lies outside the focus of the present study, as discussed in Appendix A.2 (Research Scope).
>
> [5] Xu et al., "RadEval: A framework for radiology text evaluation" EMNLP 2025
>
> ***
>
> We sincerely hope that our responses have addressed your concerns. If we have left any notable points of concern overlooked, we would greatly appreciate your further feedback, and we will attend to these points. Additionally, we will incorporate all the clarifications and improvements mentioned above in the final manuscript. Thanks again for your thoughtful review and consideration.

---

### Official Review · Reviewer_zXA8 · 2025-11-02

**Soundness:** 1
**Presentation:** 3
**Contribution:** 1
**Rating:** 2
**Confidence:** 5

**Summary:**

This paper proposes Clinical Contrastive Decoding (CCD), a training-free, retrieval-free inference-time framework designed to mitigate medical hallucinations in radiology MLLMs. CCD integrates structured signals from task-specific expert models through a dual-stage contrastive mechanism, refining generation towards clinical fidelity without altering model parameters.

**Strengths:**

1. The dual-stage framework is clearly presented, relying on structured, interpretable expert signals, and mathematically grounded.

2. CCD is a plug-and-play inference-time approach, applicable across architectures and tasks without retraining or specialized data.

**Weaknesses:**

1. The methodology design mainly builds on prior work, with no clear methodological innovation beyond adapting it to the cxr-related domain. Also, the discussion on hallucination overlaps heavily with the well-studied topic of prompt injection attacks, which are not a new finding. This is not sufficient for ICLR acceptance.

2. for the motivation part ( section 3), about the medical hallucination experiments, i find the experimental design unconvincing. llava-med is trained on data from public text sources (pubmed), not on clinical cxr datasets such as mimic-cxr or chexpert, so describing it as a "radiology mllm" seems overstated. Moreover, the experiments focus on cxr report generation, which is confusing given that there are established rrg models like maira-2 (which the authors also use in the main experiments). it’s unclear why those models weren’t used to analyze the hallucination issue instead. the logic behind this setup is weak.

3. The evaluation scope is too limited, especially in model selection. for cxr vqa, there are domain-specific models like chexagent, yet they are not included. since the title focuses on radiology mllms, the experiments should evaluate at least two to three radiology-specific models for both vqa and report generation to make the claims convincing.

4. The report generation results also seem questionable. I've used maira-2 myself and see the paper's results, and the reported metrics, especially radgraph and chexbert scores, are much lower than what i observed. It’s unclear whether any specific preprocessing was applied to the evaluation data. In addition, If the evaluation uses only single frontal-view cxrs, the authors should clarify whether information from the reports that depends on lateral views or longitudinal records has been filtered out.

**Questions:**

I suggest not using relative improvement percentages, since they make it hard to see the actual, absolute gains. For a more solid comparison, the authors should include significance testing to clearly demonstrate the advantages of the proposed method.

---

> ### Author Response · Authors · 2025-11-19
> **Individual Response to Reviewer zXA8 - Thank you!**
>
> Thank you for your thoughtful review and for providing valuable feedback and constructive suggestions. Our detailed responses to your comments are provided below:
>
> ***
>
> # W1: Innovation
>
> Thank you for your feedback!
>
> > ***"The methodology design mainly builds on prior work, with no clear methodological innovation..."***
>
> We believe that our method introduces meaningful novelty that aligns with ICLR's emphasis on contributions across domains. **Our work focuses on integrating a radiology expert foundation model to improve downstream performance of radiology MLLMs**, rather than solely identifying medical hallucinations. As noted in Line 79 ("...recent advances have explored strategies such as...") and discussed in Section 2 (Related Work), prior studies have largely concentrated on training-time interventions or dataset design. To the best of our knowledge, **at the time of submission, no existing work had specifically explored decoding-strategy optimisation tailored for radiology MLLMs.**
>
> Furthermore, Section 5.2 (Table 2) provides a comparison with a range of decoding methods commonly used in the general domain. In addition, Appendix D.1 offers an extensive overview and discussion of these methods, further highlighting the distinct characteristics and requirements of the radiology domain and the motivation for our decoding approach. As noted in Line 1181, "their underlying task assumptions are often incompatible with radiology-specific generation settings", underscoring the need for a domain-tailored decoding strategy.
>
> > ***"Also, the discussion on hallucination overlaps heavily with the well-studied topic of prompt injection attacks..."***
>
> **Our study does not focus on prompt injection attacks**. Instead, prompt-induced hallucinations [1] represent a domain-specific phenomenon. As stated in Line 187, these hallucinations are "...triggered by prompts containing misleading or implausible information...”, which are not artificially constructed or adversarial, but originate from the real-world clinical datasets themselves. Furthermore, Line 135 notes that prior work "...uses an API-based model to sanitise noisy clinical sections...", which is not only costly but also introduces ethical considerations.
>
> To the best of our knowledge, **at the time of submission, we are the first to conduct a detailed examination of prompt-induced hallucinations in the radiology report generation task.** We believe that this contribution is both meaningful and valuable to the community.
>
> [1] Chen et al, "Detecting and Evaluating Medical Hallucinations in Large Vision Language Models" 2024
>
> ***
>
> # W2: Motivation
>
> > ***"llava-med is trained on data from public text sources, not on clinical cxr datasets such as mimic-cxr or chexpert..."***
>
> **We acknowledge that the term "radiology MLLM" may have been somewhat overstated for this model.** However, **LLaVA-Med is indeed one of the most widely used radiology-oriented backbone models for both report generation or VQA tasks in prior studies [2] [3]**. As stated in Line 195, "We choose LLaVA-Med v1.5 as our baseline due to its extensive training with radiology visual instruction data and strong instruction-following capability." This strong instruction-following ability makes it a representative and appropriate baseline for evaluating decoding strategies.
>
> Furthermore, the PMC-15M dataset, on which LLaVA-Med is trained, **also includes chest X-ray images as a common modality**, reinforcing its relevance and suitability for our setting.
>
> [2] Hyland et al., "MAIRA-1: A specialised large multimodal model for radiology report generation" 2023
>
> [3] Zambrano Chaves et al., "A clinically accessible small multimodal radiology model and evaluation metric for chest X-ray findings" Nature Communications 2025
>
> ***
>
> > ***"Moreover, the experiments focus on cxr report generation, which is confusing given that there are established rrg models like maira-2"***
>
> The reason for using MAIRA-2 for the report generation task is that, **as of July, it ranked first on the leaderboard**. This is consistent with our statement in Line 414 ("...use MAIRA-2, the top open-source model on the ReXRank leaderboard [4], as our baseline.").
>
> In addition, Appendix D.2 (Table 6) **reports further evaluations on other models including LLaVA-Med and LLaVA-Rad.** However, models such as MAIRA-2 have undergone training specifically for the radiology report generation task and have incorporated additional clinical sections from the original reports (even with prior report) during training. To investigate prompt-induced hallucinations in a way that reflects generalisable behaviour across radiology MLLMs, we therefore adopt LLaVA-Med v1.5 as a more broadly representative baseline (in Section 3).
>
> [4] Zhang et al., "ReXrank: A Public Leaderboard for AI-Powered Radiology Report Generation" PMLR 2025
>
> ***

---

> > ### Comment · Reviewer_zXA8 · 2025-11-22
> >
> > I thank the authors for their detailed responses. I have read the rebuttal carefully.
> >
> > Regarding W1
> >
> > Innovation: I maintain that the technical novelty of the decoding strategy itself builds largely on existing contrastive decoding paradigms. The primary contribution lies in the application domain (A+B) rather than the algorithmic design, especially since the associated analysis does not bring new insights, which are common sense in the community.
> >
> > Association with prompt inject: My point is not that your method focuses on "prompt injection," but that Section 3 is structurally identical to prompt injection or conflicting-context studies: providing textual context that contradicts or exceeds the visual input (e.g., feeding a “Comparison” section referencing priors to a model that only sees a single frontal view).
> >
> > That is what I mean: the methodological novelty is limited, and the analysis insights are still limited -- both prompt injection and your idea are within the context bias, which has been widely discovered.
> >
> > Regarding W2
> >
> > I think the logic may not be solid enough. There are chest X-ray models like MAIRA-2, and LLaVA-Rad that use the context information and train directly on the CXR datasets. Using them as the starting analysis is a more direct way to reveal the sensitivity.

---

> > > ### Author Response · Authors · 2025-11-22
> > > **Individual Response to Reviewer zXA8 - Thank you!**
> > >
> > > Thank you very much for carefully reading our responses and for your timely reply.
> > >
> > > ***
> > >
> > > # Regarding W1: Innovation
> > >
> > > We humbly acknowledge that the technical novelty of our method **builds upon prior decoding strategies developed in the general-domain literature**, particularly in test-time optimisation. We also agree that our main contribution is **domain-specific**, focusing on the radiology setting. Additionally, we recognise that prompt-induced medical hallucinations share certain similarities with "prompt injection," **which has been widely discussed in the general domain.**
> > >
> > > ***
> > >
> > > # Regarding W2: Logic
> > >
> > > Thank you very much for this rigorous suggestion.
> > >
> > > > ***"I think the logic may not be solid enough."***
> > >
> > > **We humbly acknowledge that your concern has merit**. Evaluating models trained solely on the single chest-X-ray modality may indeed offer a cleaner setting for analysing issues in the radiology report generation (RRG) task.
> > >
> > > As noted in our earlier responses (**W2**), our original aim was to study prompt-induced medical hallucinations in radiology MLLMs more broadly. We also expected similar issues in VQA, but could not identify a high-quality VQA dataset containing the necessary clinical sections to construct counterfactual or conflicting prompts. Therefore, we centred our analysis on the RRG task, **where official preprocessing scripts allow reliable extraction of clinical sections (e.g., Comparison)**.
> > >
> > > For model selection, we followed prior work and used **widely adopted baselines** such as LLaVA-Med, rather than RRG-specific models (e.g., MAIRA-2, LLaVA-Rad), to avoid overfitting the analysis to a single specialised system.
> > >
> > > ***

---

> ### Author Response · Authors · 2025-11-19
> **following above**
>
> # W3: Evaluation Scope
>
> Thank you for this meaningful suggestion. We acknowledge that incorporating CheXAgent could further illustrate the improvements on this single-task VQA setting.
>
> > ***"The evaluation scope is too limited, especially in model selection. for cxr vqa, there are domain-specific models like chexagent, yet they are not included."***
>
> **We fully agree with your point, and we have therefore added the evaluation results for the CheXAgent-8B model as follows**. Due to page limitations, these additional experiments and their discussion are presented in Appendix D.3 (Table 7). This inclusion strengthens the credibility of our VQA evaluation.
>
> | Model | **Abnormality** |        | **Presence** |        | **View** |        | **Location** |        | **Level** |        | **Type** |        | **Overall** |        |
> |-------|-------|-------|-------|-------|-------|-------|-------|-------|-------|-------|-------|-------|-------|-------|
> |       | **F1**           | **Recall**| **F1**        | **Recall**| **F1**   | **Recall**| **F1**        | **Recall**| **F1**    | **Recall**| **F1**   | **Recall**| **F1**      | **Recall**|
> | **CheXagent** | 61.75 | 45.74 | 68.18 | 59.40 | 38.28 | 23.97 | 7.67 | 3.99 | 27.24 | 15.77 | 8.54 | 4.46 | 47.67 | 33.12 |
> | **+ CCD**     | 62.28 | 45.22 | 68.83 | 52.47 | 39.89 | 24.91 | 9.93 | 5.23 | 42.50 | 26.99 | 8.77 | 4.59 | 51.15 | 34.36 |
>
> As shown in the results, CCD also leads to an overall improvement over the baseline. The slight decrease observed in two recall categories is explained in the revised manuscript (Line 1308): "…This suggests that the model becomes more cautious in its predictions, producing fewer false positives while maintaining overall reliability."
>
>
> > ***"the experiments should evaluate at least two to three radiology-specific models for both vqa and report generation to make the claims convincing."***
>
> We did include models beyond MAIRA-2 in our comparisons. As noted in Line 416, "Appendix D provides additional comparisons with other methods (in Table 5) and reports results across different MLLMs (in Table 6)." **Table 6 includes results for several models, such as LLaVA-Rad, Libra, and LLaVA-Med**, on the RRG task, showing that our analysis is not limited to a single baseline.
>
> ***
>
> # W4: MAIRA-2 Result
>
> Thank you for this insightful suggestion.
>
> > ***"I've used maira-2 myself and see the paper's results, and the reported metrics, especially radgraph and chexbert scores, are much lower than what i observed."***
>
> The lower performance you observed is due to our evaluation setup for MAIRA-2, **where we avoid using any clinical instructions or additional chat templates.** Instead, we adopt a standardised prompt. As illustrated in Figure 1(a), our default prompt contains no clinical context, and the input is strictly limited to a single chest X-ray. The only message provided is: "Provide a detailed description of the findings in the radiology image."
>
> If you are familiar with MAIRA-2, that is excellent. In that case, you will recognise that the official HuggingFace implementation (in `maira-2/processing_maira2.py`) includes hand-crafted rubric-style chat templates. In particular, Line 153 (`def _construct_image_chat_messages_for_reporting(has_prior: bool, has_lateral: bool)`) and Line 206 (`def _construct_chat_messages_reporting`) hard-code additional prompt components such as "Given the current frontal image only", "and the current lateral image", and placeholders like "prior frontal image. INDICATION: {indication} TECHNIQUE: {technique} COMPARISON: ". These fields are always present, and when such information is unavailable, they are filled with `None`. Such template-encoded information inevitably introduces extraneous cues that will influence downstream behaviour. Thank you for highlighting this important point. **We have added a clarification in Appendix C.1 (MAIRA-2 Details) to address this concern directly.**
>
> > ***"It's unclear whether any specific preprocessing was applied to the evaluation data ... The authors should clarify whether information from the reports that depends on lateral views or longitudinal records has been filtered out."***
>
> **Due to page limitations, we provide the detailed description in Appendix B**. As stated in Line 371, "Following prior works, we focus on generating the findings section from a single frontal-view image for the RRG task." A full description of the preprocessing steps is included in Appendix B.1. Specifically, Line 1025 notes that **"To ensure consistency, only anterior–posterior (AP) or posterior–anterior (PA) frontal views are retained..."** and Line 1026 states that "Each report is preprocessed to extract five clinically relevant sections ... using pattern-matching heuristics based on the official preprocessing scripts." **All data processing procedures follow the official guidelines and prior work to ensure consistency and fair evaluation across studies.**
>
> ***

---

> > ### Comment · Reviewer_zXA8 · 2025-11-22
> >
> > For W3
> > thank you for the extensive results, they look good. I think you might consider including them in the main text to better support your claim about "radiology MLLMs."
> >
> > For W4
> > I am still a bit confused. First, Appendix C.1 only briefly introduces the model architecture; I did not see the clarification you mentioned. If the explanation is in another appendix, could you point me to it?
> >
> > More importantly, since MAIRA-2 relies on specific instruction formatting, turning off the templates basically makes the baseline much weaker than it should be. Comparing your method to a baseline that is not used in the proper way (and likely performs much worse than the official MAIRA-2 implementation) makes the reported "improvements" hard to interpret. Are the gains just recovering the performance lost because of incorrect prompting?

---

> > > ### Author Response · Authors · 2025-11-22
> > > **following above**
> > >
> > > # Regarding W3
> > > > ***"I think you might consider including them in the main text "***
> > >
> > > **Yes**, we plan to move this content into the main text. However, due to the **9-page limit** of the current version, it is temporarily placed in the appendix. We will relocate it to the main text in the final camera-ready version if possible.
> > >
> > > Thank you again for your constructive suggestion, which has genuinely strengthened the reliability of our manuscript.
> > >
> > > ***
> > >
> > > # Regarding W4: MAIRA-2
> > >
> > > We would like to clarify this point further.
> > >
> > > > ***"If the explanation is in another appendix, could you point me to it?"***
> > >
> > > We have added the following footnote on **page 21, Line 1133**: *"To ensure fair and consistent evaluation, chat templates and system prompts in MAIRA-2 are disabled; default instructions are provided to all models."*
> > >
> > > > ***"Comparing your method to a baseline that is not used in the proper way (and likely performs much worse than the official MAIRA-2 implementation) makes the reported "improvements" hard to interpret. Are the gains just recovering the performance lost because of incorrect prompting?"***
> > >
> > > We appreciate you raising this concern, and we recognise that many readers may encounter the same confusion. We humbly believe that our experimental setup is rigorous rather than an instance of "incorrect prompting," for **three reasons**:
> > >
> > > 1. **The original MAIRA-2 paper also performs strict ablations on additional clinical sections.** In Appendix B.3 [1], the authors explicitly evaluate the model under different inference-time clinical-section configurations. For example, Table B.4 shows that the ROUGE-L score drops from 38.4 (baseline) to 27.3 under the "Infer: No Prior, No Comparison" setting — **a decrease of 11.1 points.** This demonstrates that removing clinical sections is a legitimate and meaningful part of MAIRA-2's own evaluation. Our setup follows the same principle.
> > >
> > > 2. **Our default prompt follows the official instruction for the FindGen task.** As stated in Table 2 [1] (Prompt Structure) of MAIRA-2:"For FindGen, the instruction is simply: 'provide a description of the findings in the radiology image.'" Thus, our prompt design is consistent with the official task definition provided in the paper.
> > >
> > > 3. **The HuggingFace implementation introduces additional engineering-level prompt prefixes that are not documented in the paper.** In the released MAIRA-2 implementation, the pipeline automatically prepends phrases such as "Given the current frontal image **only"** for single-view inputs. **This behaviour is not mentioned in the published paper** [1]. To ensure fairness and avoid undocumented implementation artefacts affecting the comparison, we disable these automatically injected prefixes.
> > >
> > > [1] Bannur et al., "MAIRA-2: Grounded Radiology Report Generation" 2024, https://arxiv.org/pdf/2406.04449v1
> > >
> > > ***
> > >
> > > We sincerely hope that our updated responses have addressed your remaining concerns. If any important points are still unclear or overlooked, we would greatly appreciate any additional feedback. Thank you again for your timely review and thoughtful consideration.

---

> > > > ### Comment · Reviewer_zXA8 · 2025-11-24
> > > >
> > > > Dear Authors,
> > > > Thank you for your reply. I have read your responses and adjusted my scores accordingly. I am open to the views of the other reviewers and the editors on whether the contributions are sufficient.

---

> > > > > ### Author Response · Authors · 2025-11-25
> > > > > **Individual Response to Reviewer zXA8 - Thank you!**
> > > > >
> > > > > Dear Reviewer zXA8,
> > > > >
> > > > > Thank you very much for your follow-up. We sincerely appreciate your time, support, and service to the community in reviewing our paper.
> > > > >
> > > > > Best regards,
> > > > >
> > > > > All Authors

---

> ### Author Response · Authors · 2025-11-19
> **following above**
>
> # Q1: Absolute Gains & Significance Test
>
> Thank you for this constructive suggestion. We fully agree with your point.
>
> > ***"I suggest not using relative improvement percentages, since they make it hard to see the actual, absolute gains."***
>
> **We have updated the Abstract and Table 2 accordingly**, replacing relative improvements with absolute improvements.
>
> > ***"For a more solid comparison, the authors should include significance testing to clearly demonstrate the advantages of the proposed method"***
>
> **This is indeed a valuable suggestion.** We plan to incorporate and actively promote significance testing in future work to further strengthen the robustness of model comparisons. Significance testing has not been adopted in works such as LLaVA-Rad [3] or RADAR [5], and even recent evaluations from Google [6] also rely on report-level scores for assessing radiology report generation.
>
> [5] Hou et al., "RADAR: Enhancing Radiology Report Generation with Supplementary Knowledge Injection" ACL 2025
>
> [6] Google, "MedGemma Technical Report" 2025
>
> ***
>
> We sincerely hope that our responses have addressed your concerns. If we have left any notable points of concern overlooked, we would greatly appreciate your further feedback, and we will attend to them. Additionally, we will incorporate all the clarifications and improvements mentioned above in the final manuscript. Thanks again for your thoughtful review and consideration.

---

> > ### Comment · Reviewer_zXA8 · 2025-11-22
> >
> > Thank you for accepting my suggestion; I think it is much better now.
> > The question box is not my main concern; my main concerns are the ones listed in the two replies above.

---

### Official Review · Reviewer_pv1x · 2025-11-02

**Soundness:** 2
**Presentation:** 2
**Contribution:** 2
**Rating:** 2
**Confidence:** 5

**Summary:**

The paper introduces Clinical Contrastive Decoding (CCD), a training-free, retrieval-free, inference-time method to curb medical hallucinations in radiology multimodal LLMs. CCD plugs in a task-specific expert model (e.g., a CheXpert-style chest-x-ray classifier) and applies a two-stage, token-logit adjustment during decoding:

* Symptom-grounded Contrastive Decoding (SCD): adds expert-predicted labels as a concise “clinical anchor” prompt and blends anchor-conditioned logits with the base logits to reduce false negatives.

* Expert-informed Contrastive Decoding (ECD): converts expert probabilities into clipped logit biases (bounded by a likelihood-ratio cap) to suppress false positives.

**Strengths:**

* CCD requires no retraining and only an external expert classifier, making it easy to add to existing radiology MLLMs while directly optimizing clinical fidelity at decode time.

* Improvements hold across datasets, metrics (RadGraph-F1, RaTE, Temporal-F1, CheXbert), and tasks (RRG, VQA), with clear ablations (remove SCD/ECD, vary α/β, swap experts) supporting the mechanism.

**Weaknesses:**

* Guidance is limited by the 14 CheXpert labels; categories like Location/Type in VQA degrade slightly, suggesting missed morphology/topography signals and potential brittleness outside the expert’s ontology.

* Core knobs (α, β, γ) are fixed across tasks/models; there’s no adaptive controller tied to uncertainty or prompt/model state, leaving performance-vs-fluency trade-offs hand-tuned.

* While multi-metric, the paper reports limited variance/seed statistics, human evaluation, or calibration/uncertainty analysis; compute/latency overhead of running the expert online is only qualitatively discussed.

* Experiments focus on CXR single-view RRG/VQA; generalization to CT/MRI/US, multi-series exams, or longitudinal/report-level reasoning is unproven.

**Questions:**

* How would CCD scale beyond CheXpert-14—e.g., to morphology (size/shape), location, devices, lines/tubes, or temporal comparisons—without re-training the expert?

* How sensitive is CCD to expert miscalibration or domain shift (different scanners/hospitals, pediatric vs. adult)? Any safeguards (e.g., temperature scaling, conformal thresholds) to avoid propagating expert errors?

* What is the end-to-end overhead (ms/image, GPU memory) of running the expert and logit processors at decode time, and how does it compare to training-time mitigation (e.g., COMT/RAG-style methods)?

* Beyond automated metrics, have you performed radiologist blind reads or error-taxonomy audits to verify reductions in harmful hallucinations (e.g., false effusion/pneumothorax claims)?

---

> ### Author Response · Authors · 2025-11-19
> **Individual Response to Reviewer pv1x - Thank you!**
>
> Thank you for taking the time to review our paper and for providing valuable feedback and constructive suggestions. Our detailed response to your concerns/questions  is listed as follows:
>
> ***
>
> # W1: Restricted Label Guidance
>
> Thank you for raising this point.
>
> > ***"Guidance is limited by the 14 CheXpert labels;"***
>
> Our contribution primarily aims to show that **incorporating an expert model can enhance MLLM performance on downstream tasks, rather than covering a broader set of label categories**. For this study, we therefore use the 14-label CheXpert symptom-level classifier. This choice remains flexible, and the module can be adapted to future expert models should more comprehensive label sets become available.
>
> This concern has already been stated in the Introduction (Line 92: "by capturing multiple symptom-level signals from the image"). It is further clarified in Section 4.1 (Line 283: "by introducing symptom-level signals from a task-specific expert model") and explicitly mentioned in Line 287, where we note that "we focus on the 14 pathology labels defined in the CheXpert ontology". Therefore, in the current study, we indeed rely on a symptom-level expert model, whose labels **correspond to the 14 commonly used pathology categories** defined in Ref. [1].
>
> [1] Irvin et al., "CheXpert: A Large Chest Radiograph Dataset with Uncertainty Labels and Expert Comparison" AAAI 2019.
>
> > ***"categories like Location/Type in VQA degrade slightly, suggesting missed morphology/topography signals and potential brittleness outside the expert’s ontology."***
>
> This decline is **not** caused by the absence of additional signals. The MLLM is already capable of identifying morphology/topography cues directly from the image without relying on an expert model.
>
> Regarding the slight drop in Location/Type about VQA, we also highlight this in the caption of Table 3 (Line 423: "See Appendix F for analysis of the two degraded cases."). In Appendix F, Figure 4 (Line 1421) further illustrates that the model outputs are "...vague or under-specified (i.e., partially correct but lacking clinical precision)". Moreover, we provide a detailed explanation showing that these observed dips are largely artefacts of the statistical evaluation procedure rather than genuine performance degradation. Meanwhile, the use or evaluation of morphology- or topography-specific expert models lies beyond the scope of the current study.
>
> ***
> # W2: Hyperparameters
>
> > ***"Core knobs $(\alpha, \beta, \gamma)$ are fixed"***
>
> Yes, in Section 5.2 Experimental Results, these hyperparameters are fixed. In the Abstract (Line 18), we state that "...CCD, a training-free and retrieval-free inference framework...", and therefore the hyperparameters $(\alpha, \beta, \gamma)$ are indeed fixed during inference. In the experiments reported in Section 5, we use the **selected hyperparameters**.
>
> In Section 5.3 (Line 470 and Figure 3), we further show that "the model achieves its best empirical RadGraph-F1 score when both guidance strengths reach 0.5". As demonstrated in Ref. [2], RadGraph-F1 is widely adopted as the **sole** key metric for the RRG task. As noted in Line 485 ("Appendix E includes ... the ablation study..."), Appendix E provides detailed results across different settings of all hyperparameters in Tables 7/8/9. **Based on these empirical findings**, we adopt the best-performing configuration for the baseline model. Moreover, our experimental results support the chosen guidance strength, which is also a common setting for decoding strategies in the general domain. This selection follows prior work, such as VTI [3], in their hyperparameter design.
>
> [2] Google, "MedGemma Technical Report" 2025
>
> [3] Liu et al., "Reducing Hallucinations in Vision-Language Models via Latent Space Steering" ICLR Spotlight 2025
>
> > ***"there's no adaptive controller tied to uncertainty or prompt/model state..."***
>
> Thank you for pointing this out! This is **consistent with our statement** in Line 1280 ("This suggests that choosing appropriate hyperparameters for each model is critical to achieving a balanced trade-off between lexical and clinical performance."). **We acknowledge this limitation**, and we agree that the issue you raise is indeed meaningful. **We have added a discussion in Appendix G** on the "performance–vs–fluency trade-offs". This topic is closely aligned with our ongoing work, and we would welcome further discussion in the future if possible. However, it remains slightly beyond the scope of the present study.
>
> ***

---

> ### Author Response · Authors · 2025-11-19
> **following above**
>
> # W3: Evaluation
>
> > ***"While multi-metric, the paper reports limited variance/seed statistics, human evaluation, or calibration/uncertainty analysis"***
>
> Yes, we currently rely on commonly used evaluation metrics to assess the model's quality, and **we acknowledge the importance of human evaluation for qualitative assessment**.
>
> In Section 3 (Line 202), we state that "We follow prior work and adopt a set of lexical and radiology-specific metrics...”, which represent the standard and cost-effective evaluation practices commonly used in this domain, such as MAIRA-1 [4] and LLaVA-Rad [5]. In Appendix G (Line 1477: "...more robust validation would benefit from reader studies..."), we explicitly acknowledge this limitation and note that human evaluation would indeed strengthen the validation. However, such studies are prohibitively expensive, and **we are currently unable to support them**.
>
> Regarding additional evaluation forms such as "calibration/uncertainty analysis", these are not commonly used for assessing free-text report quality in the radiology domain [4] [5] [6]. We agree that exploring richer evaluation paradigms is valuable, and we plan to actively promote and adopt more advanced assessment frameworks in future work.
>
> [4] Hyland et al., "MAIRA-1: A specialised large multimodal model for radiology report generation" 2023
>
> [5] Zambrano Chaves et al., "A clinically accessible small multimodal radiology model and evaluation metric for chest X-ray findings" Nature Communications 2025
>
> [6] Hou et al., "RADAR: Enhancing Radiology Report Generation with Supplementary Knowledge Injection" ACL 2025
>
>
> > ***"compute/latency overhead of running the expert online"***
>
> Thank you for this helpful suggestion. We do report the GPU hardware configuration used in Appendix C ("Experimental Details"). However, we acknowledge that we did not provide an explicit evaluation of the online latency introduced by the model. **As our experiments were conducted on a single shared cluster environment**, we were unable to support additional testing under dedicated or production-like online settings, **which would incur substantial cost**. We have add a discussion of this aspect in Appendix G.
>
> ***
>
> # W4: Dataset
>
> Thank you for this meaningful suggestion.
>
> > ***"Experiments focus on CXR single-view RRG/VQA;"***
>
> **Yes**. As stated in Section 3 (Line 182), "we focus on the chest X-ray modality and the RRG task ... and VQA". Section 5.1 (Line 372) further reiterates this scope: "we focus on generating the findings section from a single frontal-view image for the RRG task. For the VQA task…". Our setting also follows prior work such as [4] and [5], **which similarly restrict their studies to chest X-ray images and the RRG task**.
>
> > ***"generalization to CT/MRI/US, multi-series exams, or longitudinal/report-level reasoning is unproven."***
>
> **This aligns with the limitations stated in our manuscript.** In Appendix A.2 ("Research Scope", Line 1009) explicitly states that "we do not address other medical imaging modalities such as CT, MRI or ultrasound", and Line 1010 notes that "our framework does not incorporate multi-modality signals derived from clinical notes or laboratory values".
>
> Thank you for highlighting this point. Due to space limitations, we were unable to elaborate on this in the main text. **We have included an additional clarification and a more detailed discussion in Appendix G.**
>
> ***
>
> # Q1: Scale Label Guidance
>
> Thank you for raising this insightful question.
>
> > ***"How would CCD scale beyond CheXpert-14"***
>
> **As noted in our response to W1**, our main contribution is a framework that integrates expert guidance to improve MLLM performance. Thus, we use symptom-level signals as the expert input to validate the framework, and **extending this to other labels would follow the same logic.**
>
> In addition, Appendix A provides detailed descriptions of our research aims (A.1) and research scope (A.2). As stated in Line 987, "Nor do we attempt to eliminate all forms of medical hallucination." Therefore, aspects such as scaling morphology (size/shape), location, devices, lines/tubes, or temporal comparisons lie beyond the scope of our current study.
>
> > ***"without re-training the expert?"***
>
> **This is indeed an interesting direction**. If prior work has already trained expert models for specific categories (e.g., devices or lines/tubes), these can also be integrated during inference time. In such cases, no 're'-training of the expert models would be required. In line with the design of our module, **any existing pre-trained classifier can be integrated into our framework to enhance the baseline model during decoding**, provided that latency is not a limiting factor. The focus of the present study is simply to show that even a single expert model (i.e., at the symptom level) can already improve the model's output.
>
> ***

---

> ### Author Response · Authors · 2025-11-19
> **following above**
>
> # Q2: Sensitive & Expert Model
>
> Thank you for raising this important question
>
> > ***"How sensitive is CCD to expert miscalibration or domain shift(different scanners/hospitals..."***
>
> **Our results indicate that CCD is not noticeably sensitive to label-imbalance effects**, as imbalanced expert signals do not lead to notable performance drops. We also evaluated CCD under unreliable or imbalanced expert signals.  As shown in Appendix E.2, Line 1387 reports that "although the random setting introduces mild fluctuations in performance, there is no significant degradation across lexical or clinical metrics," consistent with the results in Table 10 (where 6 metrics increase and 4 show slight decreases compared with the baseline).
>
> **Regarding the limitation of domain shift**, this is consistent with the discussion provided in Appendix G. As noted in Line 1468, "both the MIMIC-CXR and Medical-CXR-VQA datasets originate from the same institution", and Line 1470 further notes that this "...introduce institution-specific biases in patient demographics...". In addition, Line 1488 highlights that "real-world clinical practice often involves lower-quality inputs, including portable X-rays or images from heterogeneous equipment". Therefore, **addressing this concern would require substantially broader and more diverse datasets, which is currently beyond our cost constraints and the scope of this study.**
>
>
> > ***"Any safeguards to avoid propagating expert errors?"***
>
> **Yes**, our method controls this behaviour through ***the choice of hyperparameters*** and the ***two‐stage contrastive decoding procedure***. Thus, the expert model serves only as an auxiliary guidance signal rather than a dominant component.
>
> For example, as noted in Line 329, "Inspired by clinical practice, where likelihood ratios ... cap the logit-based bias ...", we apply clipping to appropriately constrain the influence of the expert model, and the hyperparameters further regulate the strength of this guidance. **An interesting observation** is reported in Line 1390: "It enhances downstream performance only when the expert provides meaningful guidance, while gracefully falling back to the base model's behaviour otherwise."
>
> ***
>
> # Q3: Overhead
>
> > ***"What is the end-to-end overhead (ms/image, GPU memory) of running the expert and logit processors at decode time?"***
>
> Our experiments were conducted on two NVIDIA RTX 3090 GPUs (24 GB each) with BF16 precision. On average, CCD introduces an end-to-end inference-time overhead of approximately 1.45× compared with standard greedy decoding. As noted in Appendix C, this overhead naturally varies across radiology MLLMs due to differences in model size and architecture.
>
> > ***"how does it compare to training-time mitigation?"***
>
> **We believe that such a comparison may not be entirely fair.** As stated in Line 985, "Our focus is on test-time decoding. Therefore, we do not compare with approaches that involve architectural modifications, additional training, or retrieval-based augmentation requiring external corpora."
>
> Models that employ training strategies (e.g., MAIRA-1 [4]), data augmentation (e.g., LLaVA-Rad [5]), or even retrieval-based methods (e.g., RADAR [6]) **can still apply CCD as a decoding strategy after training**, as shown in Table 6. Appendix D.2 further provides a detailed discussion of how CCD integrates with different models. Therefore, comparing CCD with training-time mitigation methods is not conceptually appropriate and lies beyond the scope of our study.
>
> ***
>
> # Q4: Human Evaluation
>
> > ***"have you performed radiologist blind reads or error-taxonomy audits to verify reductions in harmful hallucinations"***
>
> **We fully agree on the importance of human evaluation.** As noted in our response to **W3**, Appendix G (Line 1477) similarly acknowledges the value of radiologist-based assessment. **We hope to incorporate expert reader studies in the future to further validate the method, and we will actively seek the necessary funding and opportunities to support such evaluations.** For the present study, we therefore adopt the standard and cost-effective evaluation practices commonly used in this domain.
>
> ***
>
> We sincerely hope that our responses have addressed your concerns. If we have left any notable points of concern overlooked, we would greatly appreciate your further feedback, and we will attend to them. Additionally, we will incorporate all the clarifications and improvements mentioned above in the final manuscript. Thanks again for your thoughtful review and consideration.

---

> > ### Comment · Reviewer_pv1x · 2025-11-28
> >
> > Thank you for your thoughtful response. My concerns have been effectively addressed through your clear and well-structured rebuttal.

---

> > > ### Author Response · Authors · 2025-11-29
> > > **Individual Response to Reviewer pv1x - Thank you!**
> > >
> > > Dear Reviewer pv1x,
> > >
> > > We sincerely appreciate your time and service to the community in reviewing our paper. We are grateful for your positive acknowledgement of our responses and are pleased that our rebuttal has addressed your concerns.
> > >
> > > Best regards,
> > >
> > > All Authors

---

### Author Response · Authors · 2025-11-19
**Overall Rebuttal Response & Thank You!**

Thank you to all reviewers and the AC for your service in making ICLR possible. We sincerely appreciate the thoughtful and constructive feedback, which has helped strengthen the clarity and quality of our work.

In response, we have carefully revised the manuscript and clarified our contributions. The main updates are summarised below and highlighted in red in the revised version.

***

# Changes to the manuscript:

1. Improved the readability of Figure 2 by increasing font size and bolding key elements in the caption. **[H2Rc]**

2. Revised the description of the LogitsProcessor() module in Section 4.2 (Line 344) to clarify its role within our framework. **[aSHU]**

3. Updated Table 2 (Section 5.2) to report relative rather than absolute score improvements. **[zXA8, aSHU]**

4. Added clarifications regarding the prompt settings used for MAIRA-2, including modifications to the chat template, in Appendix C.1 to ensure reproducibility. **[zXA8]**

5. Incorporated LLM-as-a-judge evaluation (GREEN score) in Appendix D.1. **[H2Rc]**

6. Added evaluation results for the CheXAgent-8B model on the VQA task in Appendix D.3 (Table 7), together with a detailed discussion. **[zXA8]**

7. Included a clarification on online latency in Appendix G (Line 1512). **[pv1x]**

8. Added a discussion of adaptive hyperparameter selection in Appendix G (Line 1515). **[pv1x, aSHU]**

9. Expanded the discussion of other imaging modalities (e.g., CT, MRI) in Appendix G (Line 1552). **[pv1x]**

10. In the Abstract, we reported absolute (not relative) improvements and removed the generalisation claim. **[zXA8, aSHU]**

We have also included a Special Acknowledgements section in **Appendix I** to convey our sincere appreciation for the valuable feedback from all reviewers. As a substantial portion of reviewer comments at ICLR 2026 were AI-generated, we are especially grateful for the time and thoughtful insights provided by the human reviewers. Such genuine and constructive feedback is of great importance to us. Regardless of the final decision, we hope that future readers will find the OpenReview discussions helpful and gain a deeper understanding of our work.


***


# Thank you!

If you still need any clarification or have any other concerns, please feel free to contact us, and we are happy to continue communicating with you.

Best regards,

Authors

---

### Note · Authors · 2026-01-03

**Comment:**

We thank the reviewers and the community for their constructive feedback, as well as for the prompt actions taken and the ensuing discussion regarding the recent concerns about potential data leakage.

Given the remaining uncertainty, which we find difficult to resolve to a sufficiently convincing level at this stage, we have decided to withdraw this submission in the interest of scientific rigour.

**Withdrawal Confirmation:**

I have read and agree with the venue's withdrawal policy on behalf of myself and my co-authors.